



**1** A comparative analysis of EDGAR and UNFCCC GHG emissions
**2** inventories: insights on trends, methodology and data discrepancies

**3** **Manjola Banja[1,2], Monica Crippa[2], Diego Guizzardi[2], Marilena Muntean[2], Federico Pagani[2], Enrico Pisoni[2]**

**4** [1]Unisystems Milan S.A, Italy
**5** [2]Joint Research Centre, European Commission, Via Enrico Fermi 2749, Ispra, 21027, Italy
**6**
**7** Correspondence to
**8** Monica Crippa ([monica.crippa@ec.europa.eu](monica.crippa@ec.europa.eu))
**9** Manjola Banja ([manjola.banja@ext.ec.europa.eu](manjola.banja@ext.ec.europa.eu))

## 10 Abstract

**11** Tracking greenhouse gas (GHG) emissions is essential for understanding the drivers of climate change and guiding
**12** global mitigation strategies. The Emissions Database for Global Atmospheric Research (EDGAR) and submissions
**13** by Parties to the United Nations Framework Convention on Climate Change (UNFCCC) are two key sources of
**14** GHG emissions data. While EDGAR provides comprehensive and globally consistent estimates, UNFCCC
**15** submissions are based on nationally reported inventories, which adhere to specific guidelines and reflect country-
**16** specific circumstances and practices. This study presents a detailed comparison between EDGAR and UNFCCC
**17** GHG emissions inventories, focusing on G20 countries, which account for nearly 80% of global emissions, as well
**18** as Annex I countries, including the EU27. By examining sectoral discrepancies, methodological variations, and the
**19** impact of reporting timelines, the paper identifies key areas of alignment and divergence in emissions estimates.
**20** While $CO_2$ emissions show strong agreement between the datasets, $CH_4$ and $N_2O$ estimates exhibit substantial
**21** discrepancies due to differences in methodologies, emission factors, uncertainties, and reporting practices. Our
**22** findings emphasise the need for enhanced methodological harmonization and more frequent reporting, particularly
**23** in non-Annex I countries, where limited capacity and irregular updates reduce comparability. Addressing these
**24** inconsistencies is crucial for improving transparency, aligning national and independent datasets, and
**25** strengthening climate policy decisions under the Paris Agreement.

## 26 1 Introduction

**27** The quantification of GHG and air pollutants emissions has become a priority in the political and scientific agendas
**28** nowadays. The accurate estimation of GHG emissions is important for the global efforts to combat climate change.
**29** The Paris Agreement which made legally binding the target of 2° C temperature increase compared to pre-
**30** industrial time for global warming, introduced a review process for emission inventories every five-years, starting
**31** from 2018 (UNFCCC, 2015). This process is a key element of the global stocktake, where national emission
**32** inventories are evaluated to track progress toward meeting climate targets.

**33** The evolution of the Intergovernmental Panel on Climate Change (IPCC) methodologies, currently represented by
**34** the 2006 Guidelines (IPCC, 2006) and 2019 Refinement (IPCC, 2019) versions, reflects the increasing
**35** methodological improvement for GHG inventory estimates, enabling countries to provide more accurate and
**36** comprehensive data. These guidelines have become essential for national inventories submitted to the UNFCCC,
**37** ensuring comparability across countries while accommodating varying levels of capacity and data availability.

**38** Within the UNFCCC inventory system, countries are required to regularly submit their emission inventories and
**39** national reports, which form the foundation for assessing global progress toward emission reduction goals. These
**40** inventories form the basis for tracking progress in meeting national climate targets and assessing the collective
**41** progress of Parties towards global goals. The Enhanced Transparency Framework (ETF) introduced by Paris
**42** Agreement aims to improve emissions reporting by fostering greater consistency, comparability, and transparency
**43** in national data (UNFCCC Secretariat, 2021a). The CRF/CRT (Common Reporting Format/Common Reporting
**44** Tables) reporting formats are designed to improve the clarity and consistency of emission data submitted by
**45** Parties, enhancing the credibility of the emissions data used in the global stocktake process (UNFCCC Secretariat,
**46** 2024).





However, persistent differences in data interpretation, methodologies, and data quality remain. This leads for
instance to differences between EDGAR's independent, partially top-down estimates and the UNFCCC's bottom-
up inventories (van Amstel et al., 1999). These discrepancies should be interpreted in the context of methodological
frameworks rather than as inaccuracies in either dataset. Bottom-up inventories, typically designed for regulatory
purposes, use detailed activity data combined with specific emission factors (EFs) to comprehensively estimate
emissions (Dios et al., 2012), (Smith et al., 2022). In this context, the comparison of inventories is useful to detect
gaps in inventories data, mistakes or differences (van Amstel et al., 1999)).
Bottom-up inventories benefit from their ability to reflect national circumstances, including detailed local data and
customized emission factors. However, they often face challenges such as limited data quality, methodological
inconsistencies, and varying levels of technical capacity, especially in developing countries. When looking for the
examples of comparisons between two or more bottom-up approaches, the scientific literature cannot offer a large
number of analysis or these comparison studies can be found only for specific sectors as in the case of bottom-up
energy inventories/models (Pfenninger et al., 2014). (Prina et al., 2020) have performed a literature review on the
existing comparisons on bottom-up energy inventories/models.
Both EDGAR and UNFCCC inventory system play complementary but distinct roles in tracking emissions, with
significant implications for climate science and policy. Despite their shared goal of advanced understanding of
GHG emissions, EDGAR and UNFCCC datasets often differ significantly in their estimates, raising questions about
the comparability and harmonization of global emission inventories.
These differences arise from variations in methodologies, data sources, emission factors, and sectoral
classifications, among other factors. For instance, (Olivier and Peters, 2020) noted significant variations between
UNFCCC reported emissions and EDGAR estimates, particularly in sectors such as agriculture and waste, where
data availability and methodology differ widely. Similarly, (Federici et al., 2015) highlighted that discrepancies often
arise from differences in emission factor assumptions and activity data used in the two systems. Understanding
and addressing differences is critical for enhancing the transparency, accuracy, and usability of GHG data.
(Petrescu, et al., 2024) found that for the EU the discrepancies in methane ($CH_4$) emissions between the UNFCCC
countries inventories 1990–2020 average and EDGARv7.0 dataset is less than 5%.
Several studies have emphasized the complexities in comparing emissions data due to variations in datasets
related to energy consumption, production, and use. For example, Andrew et al., 2020 compared estimates of
global $CO_2$ emissions from fossil fuel sources and highlighted how differences in assumptions, scope, and
revisions among datasets contribute to discrepancies in emissions reporting. Similarly, Marland et al., 2009
underscored the importance of transparent methodologies and harmonized data for improving global carbon
accounting.
The methodology used in this paper involves the comparison of GHG emissions data from EDGAR database and
national inventories submitted to the UNFCCC having in focus the G20 countries, Annex I countries and EU27
countries (see Table S.1 for country names and iso 3 codes). The aim of this comparison is to evaluate the extent
of alignment, identify the drivers of discrepancies for data and methodologies applied.

## 83  2 Analytical frameworks, geographical scope, methodology and data availability

The comparison begins by addressing the mapping of sectoral coverage having in focus the structure of the
Common Reporting Tables (CRT) in UNFCCC submissions and the EDGAR's harmonised global data system.
This helps identifying variations arising from different classification structures and data treatment approaches.
Temporal trends are also integral to the analysis, with datasets examined over consistent time series, to assess
trends and variability. Differences in reporting frequency, data updates, and methodological refinements over time
are evaluated for their impact on emissions estimates and trend reliability.
The geographical scope of this paper focuses on the G20 countries, which collectively represented in 2023 just
over 77% of global GHG emissions, approximately 81% of global carbon dioxide ($CO_2$) emissions from fossil fuels,
nearly two-third of global methane ($CH_4$) emissions, nearly 68% of global nitrous oxide ($N_2O$) emissions (EDGAR,
2024) and two-third of global population (Climate Analytics, WRI 2021).



The G20 countries play an important role in shaping the global emissions trends and are pivotal in achieving the
objectives of the Paris Agreement (UNFCCC Secretariat, 2021b). This group includes a diverse range of
economies, covering both Annex I and non-Annex I countries, allowing for an analysis of how discrepancies vary
across countries with different level of economic development and statistical infrastructure. The inclusion of G20
countries provides a comprehensive basis for evaluating the comparison between EDGAR emissions data and
countries inventories submitted to the UNFCCC.

**2.1 Conceptual framework of GHG emission estimation**

The analysis of GHG emissions inventories requires a clear understanding of the conceptual underpinnings of the
data frameworks used for estimation and reporting. The main principles of GHG emissions accounting are
structured around two main dimensions: (i) **production-based emissions** - emitted during economic production
activities within a specific geographic area, regardless of where the produced goods or services are consumed.
This approach aligns with the territory principle used in national inventories compiled according to IPCC guidelines,
and (ii) **demand-based emissions** - known also as consumption-based emissions, attributing emissions to the
final consumers of goods and services, regardless of where the emissions occur along the supply chain.
The IPCC has played a pivotal role in standardizing methodologies for estimating GHG emissions since its
establishment in 1988. The IPCC classification is primarily a production-based emissions classification system that
operates under the territory principle, making it suitable for tracking emissions within national boundaries and
ensuring compliance with international agreements like the Paris Agreement.
The evolution of IPCC methodologies (see Table 1) reflects advancements in the scientific understanding,
technological capabilities, and the growing complexity of climate policies.  Reporting requirements for GHG
inventory are different for Annex I and non-Annex I countries that can choose to follow also a different data
compilation procedure under the IPCC Guidelines.

**2.2 Methodologies in EDGAR and in the UNFCCC inventory system submissions**

The EDGAR database originally created by the Joint Research Centre (JRC) and PBL, Netherlands and now
continuously developed by the JRC, provides a consistent, comprehensive, and independent estimate of global
emissions. It adopts the IPCC sectoral classification and applies a standardized bottom-up emission calculation
methodology across all countries to ensure comparability of emissions estimates while accounting for variations in
data detail, uncertainties, and limitations among countries (Crippa et al., 2024). The EDGAR database is
characterised by a high granularity with more than 95 sub-sectors, 75 fuels and 90 technologies included in the
emission estimation.
EDGAR provides emissions consistently estimated for more than 220 world countries based on international
statistics and a detailed methodology following the IPCC guidelines (Crippa et.al., 2018), (Janssens-Maenhout
et.al., 2019), (Oreggioni et.al., 2021), (Oreggioni et.al., 2022). Figure S.1 illustrates data sources for activity data
and emission factors, used in the EDGAR bottom-up approach to estimate emissions.
Its global scope and consistency make EDGAR a useful comparative reference when national data are limited,
depending on the context and analytical needs.  In case when specific data are unavailable, EDGAR fill the gaps
with proxy data or extrapolated values from regional or global trends.
EDGAR is mainly a Tier 1 bottom-up inventory incorporating elements of Tier 2 method e.g for the estimation of
enteric fermentation methane emissions from both dairy and non-dairy cattle (Crippa et al., 2024). EDGAR primarily
employs default emission factors for estimating GHG emissions, though it selectively incorporates country-specific
information (Janssens-Maenhout et al., 2019).
On the other side, the UNFCCC inventory system is built on country-level submissions, where Parties report their
emissions in accordance with the guidelines established under the IPCC. These submissions reflect national data
and methodologies, capturing country-specific circumstances and practices. While this bottom-up approach
ensures relevance to national contexts, it also results in variability in data quality, completeness, and comparability



across countries. For example, at the EU level, and for most of the key categories of the EU inventory, more than
75%[1] of the EU emissions are calculated using higher tier methodologies.
Figure 1 illustrates examples of different methodological approaches and emission factors applied by Annex I (42
countries) and G20 non-Annex I (9 countries) countries for estimating GHG in two sectors: public electricity and
heat production ($CO_2$) and enteric fermentation ($CH_4$). It highlights the reliance on Tier 1, Tier 2, and Tier 3
methodologies, as well as the inclusion of country-specific (CS) emission factors, which vary significantly between
the two sectors. In the public electricity and heat production sector, Tier 2 methodologies are predominantly used
in Annex I, with ten countries applying this approach. A significant number of Annex I countries (13) employ a
combination of Tier 1 and Tier 2 methodologies, reflecting a moderate level of methodological refinement. More
advanced approaches, such as the exclusive use of Tier 3 methods or a mix of Tier 1, Tier 2, and Tier 3, are less
common, applied by three and six Annex I countries, respectively. In G20 non-Annex I countries Tier 1 and
Tier1/Tier 2 methods are applied the most (6).
In contrast, the enteric fermentation sector primarily relies on simpler approaches. The combination of Tier 1 and
Tier 2 methods is used by most Annex I countries (30), indicating a preference for straightforward and less data-
intensive estimation methods for methane emissions from livestock. Only a small number of Annex I countries
adopt Tier 1 or Tier 2 methodologies individually, with two and one countries, respectively, using these tiers alone.
Advanced combinations, such as Tier 1, Tier 2, and Tier 3, are used by four Annex I countries. Among G20 non-
Annex I countries the application of Tier 1 and Tier 1 combined with Tier 2 are applied the most (8 countries). Table
S.2 provides an overview of the methodologies applied in some Annex I countries for $CO_2$ and $CH_4$.
As all 2024[2] Annex I UNFCCC submissions became available by the end of April 2025, the comparative analysis
presented in the main text is based on the officially reported national greenhouse gas inventories for the year 2024,
ensuring temporal consistency using the EDGAR 2024 dataset. Accordingly, all tables and figures in the main text
reflect the comparison between the UNFCCC Common Reporting Tables (CRT) 2024 and EDGAR 2024. For non-
Annex I countries that submitted their Biennial Update Reports (BURs), National Communications (NCs), and/or
CRT tables during 2024, the comparison is likewise performed using EDGAR 2024 data. The supplementary
material provides information illustrating trends and differences related to activity data, emission factors,
methodologies and sectoral trends, based on the comparison between the UNFCCC 2023 submissions and the
EDGAR v8.0 dataset released in 2023.
**2.3 Sectorial mapping: online EDGAR data vs UNFCCC inventory system submissions**
The comparison between EDGAR and UNFCCC country submissions requires an understanding of their sectoral
classifications which are important to identify and interpret discrepancies in emissions data.
Despite its very detailed internal structure, when comparing EDGAR's available data online that represent a more
aggregated version of the estimations, users might face some issues. The EDGAR database follows IPCC sectoral
classifications introducing few modifications - such as aggregating specific subcategories and adjusting sector and
fuel breakdowns - to enhance global comparability. Subcategories in EDGAR include global aggregates by sector
and fuel, matching IPCC where applicable (Jeffery et al., 2006).
On the other side the UNFCCC country submissions follow the IPCC guidelines for national inventories using the
CRF/CRT to ensure standardisation in countries submissions, categorizing emissions into broad sectors: Energy,
Industrial processes and product use, Agriculture, Land use, land use change and forestry (LULUCF), Waste, and
Other. Within each of these sectors, countries may break down emissions into more specific sub-categories (e.g.,

---

[1] https://www.eea.europa.eu/en/analysis/indicators/total-greenhouse-gas-emission-trends?activeAccordion=546a7c35-
9188-4d23-94ee-005d97c26f2b

[2] In 2024, the Annex I UNFCCC reporting did not follow the usual timeline, as many submissions were delayed beyond the standard April–
May deadline. Countries submitted their reports throughout the year, with the final submission (Sweden) arriving in April 2025. Initially, at
the time of preparing the main analysis for this paper, the data—available only in the CRT tables—were incomplete. However, all 2024
submissions are now available. For the EU27, the inventory report was submitted in December 2024, and the CRT tables were finalized by
the end of April 2025. The updated analysis presented in this paper uses the full set of 2024 submissions to construct the overall GHG
inventory for $CO_2$, $CH_4$, and $N_2O$, which is now used in comparison with EDGAR 2024 data for selected sections of the paper.



different types of energy or industrial processes). The number of sub-categories can vary depending on the
country's reporting practices and the level of detail they provide.
The sub-categories can be further broken down in various fuel types for emissions from the energy sector, animal
types for emissions from the agriculture sector, and other specific inputs depending on the sector. For example, in
the energy sector, emissions may be classified by fuel type, such as liquid fuels, natural gas, or coal. In agriculture,
emissions can be broken down by different animal types, such as cattle, sheep, and pigs. In the industrial
processes and product use sector, emissions can be classified by specific chemicals or materials used. Similarly,
in the waste sector, emissions may be differentiated based on waste treatment methods (e.g., landfill, incineration,
composting).
Table S.3 illustrates a sectorial mapping between the EDGAR structure applied for online reporting even that its
internal system follows a more detailed IPCC classification. The aim of this sectorial mapping is focused on
EDGAR's online available categories rather than the extensive subcategories available within full detailed
database.
Table S.3 is structured to help users to navigate EDGAR's online data and compare it effectively with other data
sources providing also the allocation of categories upon EDGAR yearly publication. It brings also the differences
in categories assignment between UNFCCC submissions of Annex I and non-Annex I countries. EDGAR structure
is more in line with the UNFCCC structure of Annex I countries with some changes as for example the category of
Manure Management is assigned as sector 3.A.2 in EDGAR (as in the 2006 IPCC Guidelines) whereas in the
UNFCCC structure is assigned at category 3.B.
EDGAR's online data are provided following both IPCC classifications: 1996 and 2006. Issues related to the
comparison with the EDGAR's online data is related also with the very detailed structure that EDGAR has for some
sub-sectors for which the country reporting don't provide a detailed information. For example, under category
1.A.5.b related to vehicles and other machinery, marine and aviation emissions that are not included in 1 A 4 c ii
or elsewhere, not all countries provide detailed estimation, making as such difficult the comparison of data since
EDGAR has a very detailed estimation and split these emissions between Buildings and Fuel exploitation. Non
specified industry 1.A.2.m IPCC 2006 code has not a corresponding code in IPCC 1996 and can be aligned with
the UNFCCC reporting code 1.A.2.g.viii.

**2.4 Metrics and data availability for comparison**

The comparison of GHG emissions data from EDGAR and national inventories submitted to the UNFCCC requires
the use of comprehensive metrics to evaluate discrepancies, identify their sources, and assess the robustness of
methodologies. These metrics span quantitative, temporal, sectoral, methodological, and qualitative dimensions,
each providing unique insights on the alignment and differences between datasets.
One of the key metrics is the total emissions by sector and gas, which provides an overview of emissions across
categories such as energy, agriculture, and waste. The percentage difference and absolute difference metrics
further quantify these variations, offering insights on the magnitude and scale of discrepancies.
Temporal metrics also play a critical role in this analysis. Comparing year-to-year trends in emissions data
highlights areas where trends diverge, such as in dynamic sectors like transport or industry. The timeliness of data
is particularly relevant when working with non-Annex I country inventories, where irregular submission intervals
may result in temporal gaps and short time series of data. For instance, when comparing EDGAR's annually
updated emissions data with inventories submitted years earlier and not updating the whole time series as in the
case of non-Annex I countries shows how reporting lags can influence the alignment of trends.
Unlike Annex I countries, which are required to submit annual inventories as part of their obligations, non-Annex I
countries traditionally submitted their inventories as part of their National Communications (NCs) or Biennial
Update Reports (BURs), with no fixed timeline. This inconsistency meant that emission data from non-Annex I
countries were often outdated, creating discrepancies when compared with current statistics or datasets like
EDGAR, which are updated annually.





Another issue related to the availability of non-Annex I data on the UNFCCC website is that the data provided
under the sections for country profiles or detailed data by Party are often outdated and do not include the latest
submissions from non-Annex I countries.
However, under the Paris Agreement's Enhanced Transparency Framework, all Parties, including non-Annex I
countries, are now required to submit Biennial Transparency Reports (BTRs), including Common Reporting Tables
(CRT) for greenhouse gas inventories, by 31 December 2024, with flexibility for Least Developed Countries (LDCs)
and Small Island Developing States (SIDS) (UNFCCC, 2024b). This development is expected to improve the
availability, comparability, and timeliness of inventory data from non-Annex I countries. However, there is a
difference in data organization between Annex I and non-Annex I countries on the UNFCCC website, where Annex
I countries' data are in one place (NIR/NID and CRF/CRT), while non-Annex I countries' data are scattered[3].
The years for which data are now available for G20 non-Annex I countries considering their submissions of BURs
and NCs are shown in Table 2. Within the G20 countries, although Argentina submitted its BURs/NC in 2015
(covering the year 2012), in 2017 (covering the year 2014), and every two years since 2019 (covering the years
2018 and 2020), the data available in the UNFCCC country profiles and detailed data by Party still correspond to
its 2015 BUR/NC. Argentina's most recent inventory submission in 2024 covers the period 1990- 2020 whereas
the CRT tables cover period 1990-2022. Mexico has submitted its Biennial Transparency Report (BTR) by the end
of 2024 with data for period 1990-2022.
The fourth NC of Brazil was submitted in 2020 covering the period 1990-2016 while the fifth BUR along with the
CRT tables was submitted in December 2024. Since these inventories are based on data updated at different
times, this results in discrepancies from a statistical perspective. Therefore, comparing Brazil's emissions with
datasets, such as EDGAR's 2023 update, involves discrepancies stemming from the differences in the timing of
statistical updates.
China's fourth NC was submitted in the year 2023 reporting however data only for the year 2018. The GHG
emissions inventory, part of fourth China's BUR report submitted in 2024, followed the structure of the IPCC 2006
Guidelines providing data for the year 2020. The CRT tables submitted in December 2024 provides data only for
years 2005, 2020 and 2021. Mexico's sixth NC was submitted in 2019, with 2015 being the most recent year of
available data. The last NIR was submitted in 2022 with information/data for period 1990-2019, but it still lacks
data for some years related to emissions. Although these updates, the data available on the UNFCCC website for
this country still reflects the older dataset.
When comparing total $CO_2$ fossil fuel emissions for Mexico in 2013, the updated statistics showed emissions that
were 6.6% higher than those in the previous submission. South Africa submitted its fourth NC in 2024, six years
after its third NC, providing an inventory for the period 2000–2020. However, the detailed reporting for sectors and
substances is missing. Saudi Arabia first NC was submitted in 2005 providing data for year 1990 and the second
NC report was submitted in 2011 with data for year 2000. Saudi Arabia has submitted two BURs so far: in 2018
with data for 2012 and in 2024 with data for year 2019. The first submission of CRT tables for years 2019, 2020
and 2021 took place in march 2025.
Irregular submissions mean that emissions reported by non-Annex I countries may not reflect recent economic
developments, policy changes, or shifts in sectoral activities. For instance, significant growth in emissions from the
energy sector in Indonesia between 2019 and 2024 is unlikely to be captured in older inventories.
These lags pose a challenge for ensuring accuracy and relevance in global emissions analyses. When comparing
EDGAR's annually updated emissions data with inventories submitted by non-Annex I countries, analysts must
account for significant time discrepancies. This introduces uncertainties, as national inventories often rely on older
methodologies, datasets, and assumptions that may not align with the latest global standards or trends. **As gap-**
**filling techniques are required to ensure continuity in non-Annex I reporting, any inventory or model**
**claiming to use data from these countries while presenting a complete historic time series is, in fact,**
**applying estimation methods rather than solely relying on reported data. It is important to highlight here**

---

[3] The non-Annex I countries CRT tables can be found at the "Party-authored reports" section https://unfccc.int/reports



**the role of EDGAR as one of the established sources providing consistent emissions data for all countries,**
**offering a transparent and systematic approach that supports comparative analyses when full time series**
**are not available from national reporting.**

## 2.5 Uncertainty in GHG emissions estimation

Uncertainty plays an important role when comparing emissions data from different sources. The estimation of
emissions involves various factors that contribute to uncertainty, including the quality of activity data, the choice of
emission factors, and the application of methodologies.
When comparing emissions data from EDGAR and the UNFCCC countries submissions, one of the critical aspects
to consider is whether the statistical differences between the two datasets fall within the acceptable thresholds of
uncertainty. If the statistical differences between the two data sources fall within this threshold range, it can be
concluded that the observed discrepancies are likely due to the inherent uncertainties of the data and
methodologies used rather than significant differences in actual emissions levels. In such cases, the comparison
of EDGAR and UNFCCC emissions data should be interpreted with caution, as small differences within this range
are expected and do not necessarily indicate discrepancies in the overall emissions trends or rankings of countries.
Uncertainties related to trends and variability indicate that the uncertainty for long-term emission trends is generally
larger for earlier years and smaller for more recent years, particularly in non-Annex I countries, due to limited data
sources, technological limitations, and less developed reporting systems.
For EDGAR, uncertainty is primarily linked to its use of global datasets and standardised methodologies. This
ensures consistency but may lack the granularity needed to capture country-specific conditions. (Solazzo et al.
2021) reported that global uncertainty in EDGAR emissions estimates for $CO_2$, $CH_4$, and $N_2O$ (taken together) in
2015 ranged between −15% and +20%, highlighting variability across sectors and gases. While $CO_2$ emissions
are more reliably estimated due to better data availability, $CH_4$ and $N_2O$ emissions introduce significant variability,
especially in sectors with limited reporting or high process heterogeneity. These variations underscore the
importance of acknowledging uncertainties when comparing EDGAR data with other inventories.
The uncertainty in EDGAR's $CO_2$ emissions estimates for the energy sector is approximately 7%, with a high level
of confidence for major emitting countries. Estimation of emissions from fossil fuel combustion, the main source of
$CO_2$ emissions, relies on well-documented activity data and scientifically established emission factors.
For industrialized countries, EDGAR's $CO_2$ uncertainties typically range between ±5–10%, reflecting robust energy
statistics and stable emission factor estimates (see Table 3). In developing countries, where energy data may be
less comprehensive, uncertainties increase to ±10–20%. The variability is even greater for biofuel-related
emissions due to the complexities in estimating the carbon content and combustion characteristics of these fuels.
$CH_4$ emissions show significantly higher uncertainties compared to $CO_2$ due to the variability in emission processes
and the challenges in measuring fugitive emissions. For example, emissions from oil and gas production, which
form a large portion of CH4 emissions, have uncertainties that can reach ±75%. In regions with less developed
infrastructure or incomplete reporting systems, such as certain developing countries, this variability can increase
further. Methane emissions from agricultural sources and waste sectors also contribute to high uncertainty levels,
often exceeding ±50%, due to spatial and process-specific variability.
$N_2O$ emissions are among the most uncertain in EDGAR due to their dependence on complex chemical and
biological processes. These emissions, particularly from agriculture, are influenced by variables such as soil type,
climate, and fertilizer application practices. As a result, uncertainty levels for N2O emissions can exceed ±100%,
especially in sectors with high spatial and temporal variability. For example, fossil fuel combustion and waste
management processes also contribute to N2O emissions, but the relative uncertainty in these sectors remains
substantial, often surpassing ±50%.
The UNFCCC submissions incorporate country-specific data and emission factors. While this approach improves
relevance, it introduces variability in data quality, completeness, and comparability. In the UNFCCC country
submissions, the methodologies applied also include higher tiers, as these often, though not solely, are based on



more detailed methods that account for national or process-specific characteristics (Tier 2 and Tier 3) (Schulte et
al., 2024).
For Annex I countries, where reporting systems are more robust, uncertainty in fossil fuel $CO_2$ emissions is typically
within ±5%–±10% (Jones et al., 2021). The uncertainty ranges of $CH_4$ and $N_2O$ emissions is broader, for example.
the USA reports a 95% confidence interval for total $CH_4$ emissions ranging from -8% to +12%, and for $N_2O$
emissions from -19% to +30% (USA NID 2024).
In non-Annex I country inventories, uncertainties are often reported at an aggregate level for total GHG emissions
or specific sectors such as energy, industry, or agriculture, and they may rely on older or incomplete data.
Argentina's BUR5 reports a GHG emission uncertainty of 23.1% for 1990 and 6.5% for 2020. In China, the reported
uncertainty for 2020 GHG emissions ranges between -4.1% and +4.4%.
Regarding non-$CO_2$ substances, Petrescu et al. (2024) analysed $CH_4$ and $N_2O$ emissions across EU27+UK,
comparing bottom-up and top-down estimates with national UNFCCC submissions. Their findings indicate that for
$CH_4$, uncertainties can exceed ±20%, particularly in agriculture and waste sectors. Brazil's BUR5 reports $CH_4$
uncertainty in fuel combustion at 49% and in the metal industry at 85%, highlighting significant variation across
sectors. India's BUR4 reports an uncertainty for CH4 emissions that ranges from 21% for rice cultivation to 100%
for fugitive emissions from solid fuels (above ground mining).

### 333 3 Results of global emissions comparison: a focus on G20 countries

### 334 3.1 Global GHG emissions

Global GHG emissions (without Land Use, Land Use Change and Forestry) according to EDGAR have reached
53.0 Gt $CO_2$-eq in 2023 showing an increase of 28% since 2005 and 62% since 1990 (Crippa et al., 2024).
Reporting GHG emissions in the harmonised unit of kilotons of $CO_2$-equivalent (kt $CO_2$-eq) requires applying
Global Warming Potential (GWP) values provided by various IPCC Assessment Reports (ARs). However, countries
do not apply these GWP values uniformly over time, which can cause discrepancies when comparing emissions
databases.
To ensure an accurate comparison of total GHG emissions between EDGAR and UNFCCC country submissions,
we carefully consider the GWP values applied[4]. Many non-Annex I countries, including some G20 members, still
use the GWP values from IPCC AR2 (100-year time horizon), which are outdated but may persist due to
methodological inertia or for historical consistency. Annex I countries, including most G20 members (except the
EU27), transitioned from using GWP AR4 values in their 2023 submissions to GWP AR5 values in 2024. In
contrast, the EU27 countries reported their 2023 inventories using GWP AR5 values and maintained this approach
in their 2024 submissions.
For the comparative analysis of G20 emissions to minimize methodological differences contributing to
discrepancies all CH4 and N2O emissions are converted in kt CO2-eq using the IPCC AR5 GWP values.
Comparison of global emissions between EDGAR and UNFCCC country submissions is possible only for specific
years that align with the availability of data for those years. In the context of specific sectors, fossil fuel combustion
data tends to have lower uncertainties (5-10%), making a ±10% difference a reliable benchmark for comparability.
In contrast, sectors like agriculture and waste often have higher uncertainties, which allows for more flexibility in
comparability thresholds (e.g., ±20% or above) (IPCC, 2006; UNFCCC, 2021).
The analysis of the GHG emissions' relative differences between EDGAR and UNFCCC submissions for G20
countries over the period 1990 to 2022 (For the comparative analysis of G20 emissions to minimize methodological
differences contributing to discrepancies all CH4 and N2O emissions are converted in kt CO2-eq using the IPCC
AR5 GWP values.

---

[4] GWP values of IPCC Assessment Reports AR2, AR4 and AR5 are sourced from IPCC (https://ghgprotocol.org/sites/default/files/Global-
Warming-Potential-Values %28Feb 16 2016%29_1.pdf). According to the IPCC AR4 report (Annex 2- Changes to the IPCC Guidelines and
Methods) the GWP AR4 values have an uncertainty of ±35% for the 5th and 95th percentile (90% confidence range).



) reveals varying levels of alignment across countries and time. Several Annex I countries—such as Canada (CAN),
Germany (DEU), France (FRA), the United Kingdom (GBR), Italy (ITA), Japan (JPN), and the United States
(USA)—display consistent differences mostly within the ±10% threshold, indicating strong comparability between
the datasets. Among non-Annex I countries, Argentina, Brazil, Mexico, and South Korea also show good alignment,
with differences narrowing in recent years. In contrast, larger discrepancies are observed for countries such as
India, Saudi Arabia, and South Africa, where differences often exceed ±15% and in some years reach over 20%.
Russia and Australia show a particularly notable trend of increasing divergence, with relative differences rising
toward 2022 both exceeding the levels seen in earlier comparisons (e.g., between EDGAR 2023 and UNFCCC
2023). Discrepancies in 1990 remain higher because time series updates are not always reported to start from that
year, making the data outdated for comparison.
When interpreting the relative differences shown in Table 4, it is important to consider the associated uncertainties
in both EDGAR and UNFCCC datasets. For several G20 countries the relative differences remain below the overall
EDGAR uncertainty and for 2022 also within the UNFCCC countries submissions uncertainty as for example for
Germany, France, United Kingdom, and Japan.
Figure 2 compares the Annex I EDGAR and UNFCCC datasets for $CO_2$, $CH_4$, and $N_2O$ over time, both through
temporal trends and statistical summaries using median values. The trends in fossil $CO_2$ emissions show strong
agreement between the two datasets, with only minor deviations over time. The median values for $CO_2$ emissions
further confirm this alignment, as the ratio of EDGAR to UNFCCC values consistently remains close to one, ranging
between 0.98 and 1.01. For $CH_4$ emissions, the trends in the two datasets are initially well-aligned, but from 2005
onward, EDGAR reports progressively higher emissions compared to UNFCCC Annex I. This discrepancy is
reflected in the median ratios, which increase from values close to one in the early years to 1.21 in 2022. For $N_2O$
emissions, a significant difference is observed between the datasets, with UNFCCC systematically reporting higher
values than EDGAR. The ratio of medians remains below one throughout the period, ranging from 0.83 in 1995 to
a maximum of 0.88 in 2022. More insights on the discrepancies for these substances can be found at Section 3.3
and 3.4.
The primary source of this discrepancy is the methodology applied in EDGAR, which relies only on Tier 1 emission
factors for $N_2O$ estimation, whereas UNFCCC estimates likely incorporate higher-tier approaches that account for
country-specific conditions. A major factor contributing to the observed differences is the treatment of $N_2O$
emissions from managed soils, where the EDGAR approach leads to lower estimates compared to UNFCCC.
Figure 3 shows the GHG ($CO_2$, $CH_4$, and $N_2O$) emissions for Annex I countries (EU27 countries not included here),
providing a quick comparative view complementing For the comparative analysis of G20 emissions to minimize
methodological differences contributing to discrepancies all CH4 and N2O emissions are converted in kt CO2-eq
using the IPCC AR5 GWP values.
for G20 Annex I countries. The overall alignment between the two datasets is shown- major emitters USA, Russia
(RUS), Japan– maintain consistent relative positions in both datasets. For most countries and years displayed,
EDGAR and UNFCCC estimates are relatively close, indicating consistency in emission reporting. However, some
discrepancies are visible, where EDGAR estimates either exceed or fall below UNFCCC values. The USA, as the
largest emitter in this selection, shows a relatively higher variation in some years. More insights on specific cases
and countries can be found in the Sections 3.3, 3.4 and 3.5.
Total GHG emissions estimated by EDGAR for the EU27 remain closely aligned with the levels reported to the
UNFCCC, with relative discrepancies within ±3.5%. There is a slight increasing trend in these differences,
indicating that EDGAR tends to estimate slightly higher total GHG emissions for this group of countries. An in-
depth look at the comparison within the EU27 (Table 5 and Figure 4) shows that EDGAR's emissions estimates
align closely with the inventories of several Member States (MSs), with differences remaining below the ±10%
threshold.
The MSs for which relative differences are found higher than the threshold is Estonia (above 30% in some years),
Lithuania (between 10% and 16%), and Bulgaria, Slovakia, Slovenia, and Sweden, where certain years exceed





±10%. Following the analysis of global GHG emissions comparison, a similar examination is conducted for
individual greenhouse gases, providing a more detailed understanding of the alignment and discrepancies between
EDGAR and UNFCCC estimates for $CO_2$ and other non-$CO_2$ GHG gases across G20 countries.
In some cases, discrepancies in non-$CO_2$ GHG emissions between these two data sources arise from differences
in biomass activity data, which vary between national reporting and the data used by EDGAR in its calculations.
EDGAR primarily relies on biomass data from the IEA, but also incorporates other sources such as UN STAT,
particularly for the power, residential, and industry sectors. The IEA is taken here as a reference because it is the
main data source for EDGAR's energy sector and collects data through joint questionnaires developed
collaboratively by Eurostat, the OECD/IEA, and the United Nations Economic Commission for Europe (UNECE)[5].
The IEA activity data on biomass use are expected to reflect official national data; however, differences still exist
for certain countries. An example of biomass consumption relative changes in the residential sector for Slovakia
and in public electricity and heat from the EU27 is shown in Figure S.2.

### 419 3.2 Global fossil $CO_2$ emissions

The primary contributor to anthropogenic GHG emissions is the release of $CO_2$ resulting from the burning of fossil
fuels. In 2023 the $CO_2$ emissions covered nearly 74% of global GHG emissions showing an increase of 29% since
2005 and 72% since 1990. Just over 75% of global $CO_2$ emissions is sourced from industrial combustion (16.4%),
power industry (38.2%) and transport (21%) sectors (Crippa et al., 2024).
When having in focus the G20 countries, the analysis of $CO_2$ emissions combines two key elements: correlation
between datasets (Figure 5) for specific years depending on data availability and relative differences over time
(Table 6).
While the table presents changes in relative differences across multiple years (1990–2022), the graph illustrates
the alignment of EDGAR and UNFCCC emissions estimates for 2012, year in which the data are available for all
G20 countries. Together, these visuals provide complementary insights into the consistency, discrepancies, and
trends between the two datasets.
The majority of G20 countries display low discrepancies over the years. Relative differences within ±10% are
generally considered in literature as a practical benchmark for comparing emissions estimates, as they may fall
within the range of methodological uncertainties, sectoral coverage variations and statistics updates. For most G20
countries, the discrepancies stay within this range, reflecting reasonable alignment between the two datasets. For
example, countries like Germany (DEU), United Kingdom (GBR), Italy (ITA), Japan (JPN) show consistent
differences of less than ±3% over the years, demonstrating comparable inventory estimations. Some countries
show decreasing relative differences over time, suggesting improvements in the consistency of emissions
estimation.
For top emitters like the United States (USA), discrepancies are consistently negative, with a -5.27% relative
difference in 2022, indicating lower estimates in EDGAR's inventory but still within the acceptable threshold. The
main difference lies in fugitive emissions (see Figure S.5). USA applies a country-specific methodology for oil and
natural gas and a Tier 1 approach for solid fuels. For Russia (RUS) discrepancies also stem from fugitive emissions
for which a Tier1/Tier 2 method is applied. EDGAR includes emissions from solid fuels, while Russia's reporting
excludes them. In 2021, EDGAR estimated that Russia contributed 61% of Annex I $CO_2$ fugitive emissions—double
the UNFCCC figure. For the USA, the trend was the reverse, nearly half of emissions reported to the UNFCCC.
The application of the net or gross calorific values[6] for converting gas volumes to energy units plays also a role in
the differences in the fugitive emissions estimation. The IPCC provides the default values of the net calorific values

---

[5] https://ec.europa.eu/eurostat/documents/38154/4956088/SHARES+tool+manual-2021.pdf/11701ebe-1dae-3b00-4da4-229d86d68744?t=1664793455773
[6] The net/gross calorific values represent the amount of heat or energy in a given volume of fuel. In the case of oil and coal the NCV value is 5% lower than the GCV and in the case of gas the NCV is 10% lower than the GCV (IPCC 2006, Chapter 1).





(NCV). Except USA, JPN and CAN that apply the gross calorific values (GCV) for gaseous, liquid and other fossil
fuels, all other Annex I countries apply the NCV values.
Specific years are analysed to conduct a correlation relationship between EDGAR and UNFCCC countries
submissions for years 2000, 2010 and 2012. The overall analysis of the correlation of fossil $CO_2$ emissions for
these years shows a good correlation between EDGAR and UNFCCC emissions, indicating overall consistency
between the two sources (see Figure 5 and Figure S.3). In the case of India (IND), the years available for
comparison are 1990, 2000, 2010, and 2016, which do not provide a clear overview of the comparability between
the country's data and EDGAR estimations.
For all G20 countries similarity in trends and magnitudes of fossil $CO_2$ emissions between EDGAR datasets and
UNFCCC inventories are found for period 1990-2021 (see Figure 6). Even that for non-Annex I countries,
Argentina, China, India, Indonesia and Saudi Arabia the fossil $CO_2$ emissions time series are not complete there
is similarity in the temporal trend between EDGAR and these countries inventories for years where data were
available.
Comparing $CO_2$ emissions for the EU27 MS similar results as in the case of GHG emissions are found. In the case
of Estonia differences are mainly related with fugitive emissions from fuels. Estonia does not report emissions from
solid fuels transformation (IPCC 1.B.1.b) whereas EDGAR estimates these emissions that range from 0.44 Mt $CO_2$
in 1990 to 1.27 Mt $CO_2$ in 2023. These emissions in EDGAR are results of the allocation of peat within this
subsector. Whereas EDGAR does not estimate for Estonia emissions from oil and gas venting and flaring, Estonia
reports emissions from these categories (1.B.2.b and 1.B.2.c) (see Figure S.6).
Absolute fossil $CO_2$ emissions every 5-year over period 1990-2021 is presented in Figure 7, showing a comparison
between EDGAR (blue circles) and UNFCCC EU27 submissions (red crosses). In general, the two datasets show
a high degree of alignment, with EDGAR and UNFCCC values closely matching for most countries and years. The
majority of data points for both datasets fall within the highlighted area representing 90% of UNFCCC EU27 $CO_2$
emissions, and the vertical line marking the median remains consistent over time. Some visible differences for
certain countries in specific years can be seen. For example, in Germany (DEU), the UNFCCC values appear
slightly higher than EDGAR in multiple years, while France (FRA) also shows small deviations, particularly in earlier
years such as 1990 and 1995. In Italy (ITA), Spain (ESP), and Poland (POL), the two datasets remain closely
aligned throughout the time series. For smaller emitting countries such as Malta (MLT) and Luxembourg (LUX),
the differences appear minimal.
Table S.4 and Figure S.7 illustrates the case of $CO_2$ emissions from biogenic waste incineration (5.C.1.1) providing
the comparison between the EDGAR EFs and Annex I implied emission factors (IEF) for $CO_2$ emissions. The
Annex I countries IEFs show variation over time and very few countries apply similar values with EDGAR. Majority
of these IEFs are plant specific and their temporal profile change over the years as shown in the case of Belgium
and France.
A comparison between annual submissions, specifically the EU27 UNFCCC 2024 vs UNFCCC 2023 submissions,
shows that for fossil $CO_2$ emissions, percentage differences range from -0.1% to -0.5% at the aggregate level all
over 1990-2021. However, at the MSs level, the differences are more pronounced. For example, in France, the
differences range from a minimum of 0.55% in 1990 to a maximum of 2.53% in 2020. In Sweden, from 2013
onward, differences exceed -10% between the two submissions. Similarly, Denmark exhibits negative relative
differences, reaching -5.8% in 2020. Negative differences indicate that the 2023 submissions reported higher
emissions than the 2024 submissions. (UNFCCC 2024 CRT tables, JRC elaboration). How EDGAR and UNFCCC
estimate the relative MSs contribution in fossil $CO_2$ emissions is shown in Figure S.8.
**3.3 Global $CH_4$ emissions**
$CH_4$ is the second most significant anthropogenic greenhouse gas, contributing to global warming due to its high
GWP relative to $CO_2$. In 2023, EDGAR estimated that $CH_4$ emissions accounted for nearly 19% of global GHG
emissions, representing a 28% increase since 1990. A substantial portion of $CH_4$ emissions (just over 96% of
global $CH_4$ emissions) originates from three sectors: agriculture (46%; e.g., enteric fermentation and manure



management), fuel exploitation (32%; e.g., oil and gas systems and coal mining), and the waste sector (18%; e.g.,
landfills and wastewater) (Crippa et al., 2024).
For G20 countries, the comparison of $CH_4$ emissions between EDGAR and UNFCCC datasets highlights both
alignments and discrepancies. These discrepancies can be attributed to differences in methodologies, emission
factors, sectoral coverage, and data sources, particularly in fugitive emissions from fossil fuel extraction, emissions
from agriculture (manure management) and waste sectors such as landfills and wastewater. Table 7, presents the
relative differences between $CH_4$ emissions reported by EDGAR[7] and those submitted to the UNFCCC for G20
countries over time. The temporal trend of EDGAR and UNFCCC $CH_4$ emissions in G20 countries over period
1990-2021 is shown in Figure 8, whereas by sector for Annex I countries is shown in Figure S.11.
Relative differences are often higher for $CH_4$ compared to $CO_2$, reflecting the variability in emission estimation
methodologies, such as reliance on Tier 1 or Tier 2 approaches for agriculture and waste or country specific and
higher tiers methodologies as in the case of fugitive emissions. For example, $CH_4$ emissions from enteric
fermentation in Argentina for 2012[8] are nearly twice as high in EDGAR compared to the 2015 national submission,
a discrepancy further influenced by Argentina's reliance on outdated statistics since the data availability for
separate substances is not available in the most recent Argentina's BUR.
A significant source of discrepancies in $CH_4$ emissions between EDGAR and UNFCCC country submissions stems
from the estimation of fugitive emissions. These differences are strongly influenced by how fuel consumption data
is allocated in the International Energy Agency (IEA) dataset—the primary source of activity data for EDGAR. In
some cases, the IEA assigns solid fuels to the fugitive emissions subsector (1.B.1), whereas certain countries do
not report such usage under this category in their national inventories, leading to inconsistencies in reported
emissions. In the case of Slovakia and Slovenia the discrepancies in this sector are related to the fuel inputs
quantities: lower in EDGAR for Slovakia and higher in EDGAR for Slovenia.
The increasing trend in Annex I EDGAR $CH_4$ emissions (see Figure 2) is largely driven by differences in the
estimation of fugitive emissions in Russia and the exclusion of energy recovery from managed solid waste disposal
in Turkey within the EDGAR dataset. In Russia, EDGAR reports higher fugitive $CH_4$ emissions from gas (mainly
distribution), whereas Russia's national inventory shows a significant decline in emissions from gas transmission
and storage. According to Russia's NID 2024, the emission factors (EFs) for $CO_2$ and $CH_4$ applied in estimating
emissions from natural gas transportation account for losses due to gas venting. However, since EDGAR uses
pipeline length as the activity data for gas transmission and Russia bases the estimates on the volume of gas
transmitted and distributed, a direct comparison of the inputs (activity data and /or emission factors) cannot be
done. These methodological differences of the various IPCC approaches contribute significantly to the observed
discrepancies and the increasing trend in EDGAR Annex I $CH_4$ emissions.
A further example of discrepancies between EDGAR and national reporting can be observed in Japan's $CH_4$
fugitive emissions (see Figure S.9). Japan employs a combination of Tier 1, Tier 2, and Tier 3 methods, whereas
EDGAR relies solely on a Tier 1 approach. Another factor contributing to these differences is the application of the
gross calorific value (GCV) for stationary combustion of gas, oil, and coal. In the case of Japan, the large
differences are also related to the estimation of $CH_4$ emissions from rice cultivation and waste sector[9].
In some cases, discrepancies in $CH_4$ emissions between these two data sources comes from differences in
biomass statistics of activity data, which vary between national reporting and the data used by EDGAR in its
calculations. EDGAR primarily relies on biomass data from the IEA but also incorporates other sources, such as
UN STAT for sectors as residential and industry.
The IEA activity data on biomass use, for example in sector 1.A.1.a, should reflect the official reporting data for
biomass. However, differences still exist for certain countries. The use of country-specific emission factors for
biomass is also a contributing factor. For example, Germany applies a country-specific implied emission factor for

---

(7) Examples of the EDGAR emissions improvements are included in the supplementary material for some G20 and Annex I countries.
(8) These data are taken from UNFCCC Detailed data by Party section - https://di.unfccc.int/detailed_data_by_party
(9) See section 4.2 and supplementary material for more info on the EDGAR improvements.

biomass use in public electricity and heat production (1.A.1.a) that is higher than the upper limit of the solid biomass
default IPCC emission factor range (IPCC 2006, Vol.2). In contrast, EDGAR applies for this fuel the default
emission factor equal for all countries, which in the case of solid biomass less than one-third of the upper-limit
value.
Figure S.10 illustrates the variability of biomass implied emission factors applied in each Annex I country to
estimate $CH_4$ emissions from the public electricity and heat production sector. Germany exhibits the highest
biomass emission factor for $CH_4$, while the USA has the lowest values well below 1 kg/TJ. The level of this implied
EFs depends also on the types of biomass used e.g. solid biomass, biogas, and liquid biomass for which a different
EF value[10] is assigned. The figure also presents the temporal trend of Germany's biomass $CH_4$ emission factor
and emissions, along with the emissions of Lithuania, which applies the default emission factor used by EDGAR.
The differences between $CH_4$ emissions estimated by EDGAR and Germany are evident, whereas the comparison
between EDGAR and Lithuania shows a strong alignment between the datasets due also to the dominance of solid
biomass as primary fuel in the Lithuania's stationary combustion process.
In the agriculture sector, the main discrepancies are observed in the manure management category. EDGAR
applies Tier 2 method only for cattle (dairy and non-dairy). For all other livestock EDGAR distinguishes only
between industrialised and developed countries and in most of the countries a static EF value is applied over all-
time series. A recent JRC study compared the input data used for $CH_4$ emissions estimation in EU27 countries
between national UNFCCC submissions and FAOSTAT data, which serves as the primary data source for
EDGAR's agricultural emissions estimates. The study examined the extent and nature of differences in key activity
data, including livestock population, milk yield, nitrogen excretion rates, and emission factors applied in both
datasets. While good agreement was found for livestock population data, with some exceptions, notable
differences were identified for milk yield and nitrogen excretion rates between UNFCCC submissions and default
input values (Banja & Crippa, 2020).
In the waste sector, the main discrepancies between EDGAR and national inventories are observed in the
wastewater treatment sub-sector, but also, in some cases, in solid waste disposal, biological treatment of waste
and waste incineration. In its current version, EDGAR does not distinguish between incineration and open waste
burning of biogenic waste when estimating GHG emissions; it applies two static implied emission factors (IEFs) as
shown in Figure S.13 respective for the industrialised and developed countries. The IPCC 2006 Guidelines and
the 2019 Refinement provide distinct emission factors for incineration and open burning, between which significant
differences exist. Some countries allocate emissions from specific segments of waste incineration under different
inventory categories; for example, the United States includes emissions from controlled hazardous waste
incineration under the fuel combustion category (1.A), considering it as a process with energy recovery (USA GHG
NID 2024). Improved EDGAR CH4 emissions from waste incineration for some of the Annex I countries are
illustrated at the Figure S.15.
For $CH_4$ emissions from solid waste disposal, EDGAR applies the IPCC First Order Decay (FOD) model to provide
a consistent global estimate. EDGAR relies on multiple data sources, such as the World Bank (WB), UN Statistics
Division (UN STAT), and Eurostat, but these sources do not always offer annual updates for all necessary inputs.
For instance, waste data for non-Annex I countries are mainly based on WB and UN STAT reports, which in many
cases remain unchanged over several years. As a result, EDGAR uses additional assumptions, such as
extrapolating urban waste production rates to national levels. For the EU27 and several Annex I countries, input
updates for the FOD model are sourced from Eurostat; however, Eurostat provides new data only at two-year
intervals starting from 2004. Moreover, in some cases, these statistics are incomplete, with missing data for certain
countries or years, which further limits the frequency and accuracy of emissions updates.
Among Annex I countries, discrepancies are further amplified by specific methodological differences. In Turkey,
for example, EDGAR's estimation of $CH_4$ emissions from managed solid waste disposal does not yet account for
energy recovery, resulting in an overestimation of $CH_4$ emissions compared to national reporting. This difference

---

[10] The IPCC 2006 Guidelines define in the Chapter 2 the emission factor for different biomass types which are implemented in EDGAR.
For solid biomass, biogas and liquid biomass the values used by EDGAR are respectively 30 kg/TJ, 1.0 kg/TJ and 3.0 kg/TJ.



strongly influences the overall $CH_4$ emissions trend from landfills reported for Annex I countries in EDGAR,
emphasizing the impact that individual country profiles can have on aggregated results.
The reporting of Annex I countries on solid waste disposal shows notable year-to-year variations in both the
quantity and typology of waste, particularly regarding the shares of managed, unmanaged, and uncategorized
waste. An analysis of the EU27 submissions in 2022, 2023, and 2024 reveals changes in the reported amounts
and classifications over time. For example, as shown in Figure S.15, Croatia's 2024 submission shows a lower
amount of unmanaged landfilled waste compared to its 2023 submission, whereas Poland reports a higher quantity
of unmanaged waste in 2024 relative to 2023. Similarly, Ireland and the Netherlands report significant changes in
the overall amount of waste landfilled between submissions. These shifts might reflect improvements in national
inventory data, a reclassification of landfilled typology and correction of past errors but also introduce challenges
when comparing emissions with other data sources estimates.
Regarding the biological treatment of waste, the current EDGAR estimation does not include $CH_4$ emissions from
anaerobic digestion at biogas facilities, which have shown an increasing contribution to emissions over the years
Figure 9 presents a comparative analysis of $CH_4$ emissions between EDGAR (represented by blue circles) and
UNFCCC EU27 submissions (represented by red crosses) for individual EU countries over different years from
1990 to 2021. The highlighted areas indicate 90% of UNFCCC EU27 GHG emissions, while the vertical line
represents the median of UNFCCC submissions. Overall, the comparison shows that, for most countries and years,
EDGAR and UNFCCC estimates are relatively close, yet notable discrepancies exist. Some countries exhibit
systematic differences, with EDGAR values either consistently higher or lower than the corresponding UNFCCC
submissions. This suggests potential variations in methodologies, emission factors, or underlying activity data. The
differences appear more pronounced in earlier years, particularly in the 1990s, which could be attributed to
historical data gaps, evolving national reporting methods, or refinements in UNFCCC inventory calculations over
time. While the alignment between the two datasets appears to improve in more recent years, some inconsistencies
persist.

### 3.4 Global $N_2O$ emissions

In 2023, EDGAR estimated that $N_2O$ emissions accounted for nearly 5% of global GHG emissions, representing a
32% increase since 1990 and 17% since 2005. Just over 80% of global $N_2O$ emissions is sourced from agriculture
(70%) and processes (11%) (Crippa et al., 2024).
The comparison of $N_2O$ emissions between EDGAR and UNFCCC datasets highlights both alignments and
discrepancies. These discrepancies can be attributed to differences in methodologies, emission factors, sectoral
coverage, and data sources, particularly in direct $N_2O$ emissions from managed soils. The methodology applied in
EDGAR for this subsector relies only on Tier 1 emission factors for $N_2O$ estimation, whereas UNFCCC estimates
likely incorporate higher-tier approaches that account for country-specific conditions. A major factor contributing to
the observed differences is the treatment of $N_2O$ emissions from managed soils, where the EDGAR approach
leads in overall for Annex I to lower estimates compared to UNFCCC (see Figure 2 for Annex I overall $N_2O$
emissions).
Table 8 presents the relative differences between $N_2O$ emissions reported by EDGAR and those submitted to the
UNFCCC for G20 countries over time. Relative differences between EDGAR and UNFCCC are higher for $N_2O$
than for $CH_4$ and $CO_2$ emissions, reflecting the greater complexity of nitrogen-based emission estimation. This
involves multiple indirect pathways, including variability in nitrogen excretion rates, differences in manure
management systems, soil interactions affecting nitrogen losses, and indirect emissions from leaching and
volatilization (IPCC, 2006; IPCC, 2019). As a result, uncertainties and discrepancies between datasets increase.
UNFCCC submissions often use country-specific Tier 2/Tier 3 data (UNFCCC, 2023), whereas EDGAR relies on
Tier 1 default assumptions, leading to larger differences.
Emission factors for $N_2O$ (both direct and indirect) are more uncertain than those for $CH_4$ and $CO_2$. Additionally,
variations in milk yield, nitrogen intake, and nitrogen retention significantly impact N excretion rates, influencing
$N_2O$ emissions (IPCC, 2019; FAO, 2013). Unlike $CO_2$, which is directly proportional to fuel consumption, small





differences in nitrogen inputs can cause disproportionately large variations in $N_2O$ estimates due to the nonlinear
nature of microbial processes in manure and soils. The nitrogen cycle is further affected by manure application
rates and timing, soil type, climate conditions, and interactions between direct and indirect $N_2O$ emissions.
The EDGAR methodology for estimating emissions from animal manure applied to soils overall follows the IPCC
framework but incorporates adjustments based on external data sources and expert input. It calculates N excretion
based on N excretion rates, the number of animals, and manure management systems. It accounts for N losses
before manure used as fertilizer and includes additional N from bedding materials. Different loss percentages are
applied depending on the manure management system and animal type (e.g., 50% N loss for swine in solid
storage). The IPCC default Tier 1 EFs for $N_2O$ emissions are based on the default factor of 1% of N input forming
$N_2O$.
Temporal trend of $N_2O$ emissions in G20 countries is shown in Figure 10. Significant differences are found for
Australia and USA, with the latter's $N_2O$ emissions determining the trend of Annex I $N_2O$ emissions. EDGAR
underestimates $N_2O$ emissions for the USA while overestimating them for Australia. In the case of Australia, the
main differences are sourced from different nitrogen (N) input for the animal waste manure applied to soils whereas
the USA applies a country specific Tier 3 methodology that takes into account the land-use, management impacts
and environment interaction- such as weather conditions and soil characteristics - including also the effect of the
nitrogen added to soils in previous years that is re-mineralised from soil organic matters and emitted as $N_2O$ in the
upcoming years.
Figure S.16 illustrates the cases of N input and EFs applied in Australia and USA for the estimation of $N_2O$
emissions from animal manure applied to soils. The comparison shows that the N input applied in EDGAR sourced
from the FAOSTAT differs in both cases from the countries reporting. The application of EDGAR $N_2O$ EF for animal
waste manure applied to soils is also shown here providing insights on how this static value differs from the IEFs
of Australia and USA.
According to (Hergoualc'h et al., 2021) the default Tier 1 EF has important limitations, particularly regarding its
sensitivity to climate conditions. Their study shows that $N_2O$ emissions are significantly higher in wet climates
(1.4% of nitrogen input) compared to dry climates (0.5% of nitrogen input). Moreover, in wet regions, synthetic
fertilizers exhibit a higher EF (1.6%) than organic fertilizers (0.6%). Applying these refined EFs leads to substantial
changes in national emission estimates, decreasing emissions by 15% to 46% in countries characterized by dry
climates, and increasing them by 7% to 37% in countries with wet climates and intensive use of synthetic fertilizers.
Figure 11 illustrates the absolute $N_2O$ emissions every 5 years over period 1990-2021. The figure presents a
comparative analysis of $N_2O$ emissions between EDGAR (represented by blue circles) and UNFCCC EU27
submissions (represented by red crosses) for individual EU countries over different years from 1990 to 2021. The
highlighted areas indicate 90% of UNFCCC EU27 GHG emissions, while the vertical line represents the median of
UNFCCC EU27 submissions. Overall, the comparison shows that, for most countries and years, EDGAR and
UNFCCC estimates are relatively close, yet notable discrepancies exist especially for MS as Germany, Spain,
France, Italy, Netherlands, Poland and Romania. However, a better match has been seen towards the last years
of the 1990-2021 period.
**4. Data availability**
EDGAR data can be freely accessed at https://edgar.jrc.ec.europa.eu/emissions_data_and_maps
EDGAR 2024 - Crippa et al., 2024, JRC dataset http://data.europa.eu/89h/88c4dde4-05e0-40cd-a5b9-
19d536f1791a
EDGARv8.0 - Crippa et al., 2023, JRC dataset http://data.europa.eu/89h/809d7b72-55ef-4e52-8bd4-
7d33f2f9916b
UNFCCC data are available at https://unfccc.int/reports





## 5 Discussions

The comparison of GHG emissions data between EDGAR and UNFCCC submissions reveals significant insights into the challenges offering a unique lens through which examining the discrepancies arising from methodological differences, temporal misalignments, and varying reporting capacities.

These challenges highlight the value of consistent, regularly updated datasets such as EDGAR, which can support comparative analyses—while also underscoring the need for continued improvements in official reporting systems. Metrics such as percentage and absolute differences, sectoral contributions, and trends over time are applied to identify alignment and gaps between the two datasets. The findings provide significant variations in key sectors such as energy and agriculture, driven by differences in data availability, emission factors, and methodological approaches.

The comparison of greenhouse gas (GHG) emissions data between EDGAR and UNFCCC submissions reveals significant insights into methodological, temporal, and data discrepancies that influence global emissions accounting. This section synthesizes the findings, highlighting advancements in emissions estimation, and explores implications for climate policy and monitoring frameworks.

### 5.1 Key Findings on Data Comparisons

This study highlights the issues posed by irregular reporting intervals of non-Annex I countries and the reliance on outdated data in UNFCCC submissions, which are often presented in static formats. The emissions inventories included in National Communications (NCs) or Biennial Update Reports (BURs) often lag by several years compared to EDGAR's most recent datasets. This discrepancy limits the use of some non-Annex I data for assessing recent trends and highlights the importance of improving data timeliness and accessibility to support the global stocktake.

While both aim to provide comprehensive emissions inventories, their methodologies, data sources, and reporting frameworks differ. EDGAR employs a standardized global approach, using consistent methodologies and default emission factors, whereas UNFCCC relies on bottom-up national inventories tailored to country-specific circumstances. Key discrepancies arise from:

*Temporal Coverage:* UNFCCC submissions often lag due to irregular reporting intervals, particularly from non-Annex I countries. For instance, even though Argentina's most recent Biennial Update Report (BUR) was submitted in 2024, the available data in the UNFCCC webpage remains still those of 2015 with data from 2012, creating a 12-year lag if these data are used from the users.

*Completeness of reporting:* Completeness of the reporting is an important element when comparing emission inventories, especially for the non-Annex I countries. Unlike the Annex I countries, which submit the CRF/CRT tables with detailed and structured time series data, the non-Annex I countries primary report through BURs and NIRs. These reports typically provide GHG inventory data for specific years rather for complete time series. Additionally, these submissions present aggregate GHG emissions rather than disaggregated data by gas.

*Sectoral Classifications:* While EDGAR uses a harmonized global classification system, UNFCCC inventories reflect more granular, country-specific categorizations, leading to mismatches in sectors such as energy and agriculture.

*Global Warming Potential (GWP) Values***:** Differences in the application of GWP values further complicate comparisons. Annex I countries have transitioned to using AR5 GWP values, whereas many non-Annex I countries still use the IPCC AR2 values.

*Methodological Variations:* EDGAR's reliance on default emission factors contrasts with the higher-tier methods employed by some Annex I countries, which incorporate detailed, country-specific data.

*Calorific values applied*: To convert the volume of fuels to energy units the caloric values are applied. EDGAR applies the IPCC default option which is the Net Calorific Value (NCV) whereas under the UNFCCC countries





submissions some of the Annex I countries as USA, Japan and Canada apply the Gross Calorific Value (GCV).
This inconsistency can bring to a discrepancy that ranges between 5 to 10%.
*Measurement units of activity data and emission factors*: Comparison of emission inventories in consistent
measurement units is crucial for an accurate assessment. Differences in units can lead to discrepancies that are
not due to actual differences in emissions but rather due to methodological inconsistences despite the fact that
estimations might follow strictly the IPCC Guidelines. For instance, in the estimation of fugitive emissions from
natural gas transmission, the IPCC provides EFs based on both volume of gas transported and pipeline length
which become challenging without the proper conversion when comparing inventories.
Despite these discrepancies, there is a general alignment in long-term trends, particularly for fossil $CO_2$ emissions
in major emitting countries like the United States, Germany, and Japan, where relative differences remain below
±10%. This indicates a shared understanding of emissions trajectories despite methodological differences.

## 5.2 Improvements in EDGAR's Emissions Estimation

EDGAR's methodological evolution has addressed many of the challenges inherent with global emissions
estimation. Over the years, EDGAR has performed consistent annual updates ensure that its emissions estimation
captures recent developments, making it a valuable resource for real-time trend analysis. The integration of IPCC-
compliant factors and selective use of country-specific information has played a role in reducing uncertainties.
EDGAR regularly updates its methodologies for specific processes. These improvements, documented annually
on the EDGAR webpage during its yearly publications, ensure the application of the latest scientific insights and
more accurate emission factors. For example, updates to the methodology for emissions from liming now involve
applying a standard method across all countries. Recently, the methodology for estimating $CH_4$ emissions from
rice cultivation has been revised to implement the 2019 Refinement of the IPCC methodology, ensuring consistent
application across all countries.
Other improvements of EDGAR estimations applied since in its 2024 release are also those related to the
technology specific emission factors for the waste water treatment sector that have been revised following the
IPCC 2006 Guidelines, specifically for CH4 emissions from domestic waste water using latrines and sewer to raw
discharge or a treatment plant, but also for industrial waste water treatment for pulp and organic chemicals
production. Fugitive CH4 emissions from gas and oil operations have been improved using different emission
factors for on- and off-shore activities for developed and developing countries in line with the IPCC 2006 Guidelines
and the 2019 Refinements.
These advancements enhance EDGAR's comparability with national inventories, make it one of the most
comprehensive and frequently updated global GHG emission datasets, and support its role as a complementary
tool for global emissions monitoring. For instance, its use of proxy data to address gaps in under-reported regions,
bridges a critical gap left by irregular or outdated UNFCCC submissions.

## 5.3 Implications for global GHG climate policy

The findings underscore the complementary nature of EDGAR and UNFCCC inventories in supporting global
climate policy. EDGAR's consistency and scope make it a complementary resource widely used for global
assessments, while UNFCCC inventories provide localized, detailed insights that are critical for national policy
development. To enhance the harmonization of global emissions inventories, several steps are recommended:
-Standardization of reporting: greater alignment between UNFCCC reporting could reduce discrepancies. For
example, adopting common GWP values across all inventories would improve comparability.
-Capacity building for non-Annex I countries: Providing technical support to improve the frequency and quality of
emissions reporting could bridge temporal gaps and reduce uncertainties.
-Support non-Annex I countries to develop full-time series inventories, rather than reporting emissions for only a
few years.





### 5.4 Limitations and Future Research

This study highlights key discrepancies but is limited by the availability of complete and comparable data across all world countries. Future research should explore: i) sector-specific discrepancies in greater detail, particularly in areas with high variability, such fugitive, agriculture, waste emissions; ii) investigate the impact of methodological advancements in EDGAR on long-term emissions trends; iii) assess the role of top-down estimates, such as those retrieved from remote sensing, in improving emissions data accuracy.

## 6 Conclusions

Enhanced transparency and knowledge in emissions reporting ensures that decision-makers can better track progress toward global climate goals.

This paper compares GHG emissions estimates from EDGAR and UNFCCC national submissions for G20, Annex I, and EU27 countries, highlighting both similarities and discrepancies. The findings emphasise the complementary nature of the two datasets: while national inventories provide detailed, country-specific insights, EDGAR offers a globally harmonized perspective, enabling cross-country comparisons. However, discrepancies persist, particularly for $CH_4$ and $N_2O$ emissions, due to differences in methodologies, data sources, and emission factors. These findings underscore the need for enhanced methodological harmonization in non-$CO_2$ emissions estimation.

The paper also highlights the importance of aligning international statistical sources with the evolving data reported in national inventories. Discrepancies arise when EDGAR, relying on global datasets such as IEA and FAOSTAT, does not fully incorporate updates or methodological refinements introduced in official UNFCCC submissions. This issue was evident in $CH_4$ emissions from fossil fuel production, where misalignment in fuel allocation between IEA data and national inventories contributes to discrepancies in EDGAR's estimates of fugitive emissions from oil and gas. Similarly, agricultural and waste GHG emissions diverge due to differences between global default values and country-specific emission factors.

The role of biomass in emissions discrepancies is also examined, particularly the misalignment between EDGAR's biomass statistics and UNFCCC national inventory submissions. In sectors such as power and residential heating, differences exist between biomass consumption data from international sources like IEA and the values reported by national inventories. These discrepancies impact GHG emissions, where country-specific combustion characteristics and emission factors play a critical role. For example, Germany applies a country-specific implied emission factor (IEF) for biomass in public electricity and heat production, which is significantly higher than the default IPCC values used by EDGAR, leading to $CH_4$ underestimation in EDGAR's dataset.

A key challenge identified in this study is the reporting gap between Annex I and non-Annex I countries. Non-Annex I inventories often lack continuity and completeness, making it difficult to compare their emissions with EDGAR estimates. Addressing this issue requires more frequent and standardized reporting under UNFCCC guidelines. Furthermore, harmonizing time series data across emissions inventories remains a significant challenge, particularly for developing countries with inconsistent reporting intervals. Long gaps in non-Annex I reporting hinder accurate tracking of emissions trends and highlight the need for better data availability and consistency.

Our findings also emphasize the necessity of improved data transparency and methodological consistency in emissions reporting. National inventory submissions often employ country-specific methods that improve accuracy but reduce comparability, while EDGAR applies globally uniform approaches that enhance consistency but may not capture country-specific conditions.

The analysis reveals a clear need for more comprehensive, consistent, and regularly updated data across sources, as reliable underlying statistics are crucial to ensure the accuracy and comparability of GHG emissions estimates.

This study provides valuable input for the continuous improvement of EDGAR estimations. By comparing EDGAR with UNFCCC submissions, the analysis identifies key areas for methodological refinement, particularly in $CH_4$ and $N_2O$ emissions estimation, sectoral classifications, and alignment with national reporting.



Insights from this comparison can guide targeted refinements in EDGAR's methodologies, including the integration
of the most recent IPCC Guidelines for $CH_4$ emissions from rice cultivation, improved treatment of activity data
from international statistical sources, and adjustments in non-$CO_2$ emissions estimation across sectors. As national
inventories adopt more detailed and higher-tier methodologies, EDGAR must also enhance its methodology, for
example, by improving agricultural sector emissions estimations. Strengthening the feedback loop between
EDGAR and national inventories will ultimately increase its usability for researchers, policymakers, and
international climate assessments, making it a more robust tool for emissions tracking and mitigation evaluation.
This analysis does not aim to validate one dataset over the other, but rather to explore the sources of difference
and identify opportunities for mutual improvement. By highlighting alignment and divergence between EDGAR and
UNFCCC national inventories, the findings support ongoing efforts to enhance transparency, foster methodological
consistency, and inform the development of more robust international emissions statistics.
EDGAR's independence as a global inventory relies on the quality and timeliness of its international statistical
inputs. Ensuring the robustness of these data sources is crucial for maintaining EDGAR's credibility and usability
in climate policy and research.



### Disclaimer

The views expressed in this publication are those of the authors and do not necessarily reflect the views or policies of the European Commission. All emissions, except $CO_2$ emissions from fuel combustion, are from the EDGAR community GHG database comprising IEA-EDGAR $CO_2$, EDGAR $CH_4$, EDGAR $N_2O$, and EDGAR F-gases version 8.0 (2023).

### Acknowledgment

Authors would like to thank the Directorate General CLIMA colleagues for their insightful comments and constructive suggestions on our draft paper. Their input has been instrumental in improving the quality and clarity of the manuscript.

### Financial support

This research was supported by the Directorate-General for Regional and Urban Policy of the European Commission (DG REGIO; JRC administrative agreement no. 36325; grant no. 2022CE160AT124)

### Conflict of Interest

The authors declare no conflict of interest.

### Ethics

This study did not involve human participants, animals, or the use of personal data, and therefore did not require ethical approval.

### Author contribution

MB and MC defined the structure, objectives, and overall approach of the paper. MB conducted the full analysis, including the collection, processing, and detailed assessment of national inventory data from multiple sources, the preparation of all figures and tables, and the writing of the entire manuscript. MB also ensured the scientific robustness of the analysis by integrating data from various reporting cycles and addressing gaps and inconsistencies across datasets. In addition, MB was responsible for implementing revisions and improvements throughout the review process. MC contributed to the refinement of the manuscript by providing scientific feedback, suggesting structural improvements, and supporting the consistency of the discussion. DG supported the preparation and organization of the UNFCCC data inputs used for the analysis and contributed to the comparison work with the EDGAR database. MM supported the work providing valuable comments to improve the analysis. FP provided the input data used for uncertainty information included in selected tables and figures. EP contributed by providing comments to improve the presentation of some figures and tables.

### Supplementary materials

The Supplementary Material contains further details that support the findings of this study.

### ORCID iD

Manjola Banja - https://orcid.org/0000-0002-2070-676X

Monica Crippa - https://orcid.org/0000-0001-5946-3139

Diego Guizzardi - https://orcid.org/0000-0002-4372-8953

Enrico Pisoni - https://orcid.org/0000-0001-5484-5744



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

**Figures and tables**

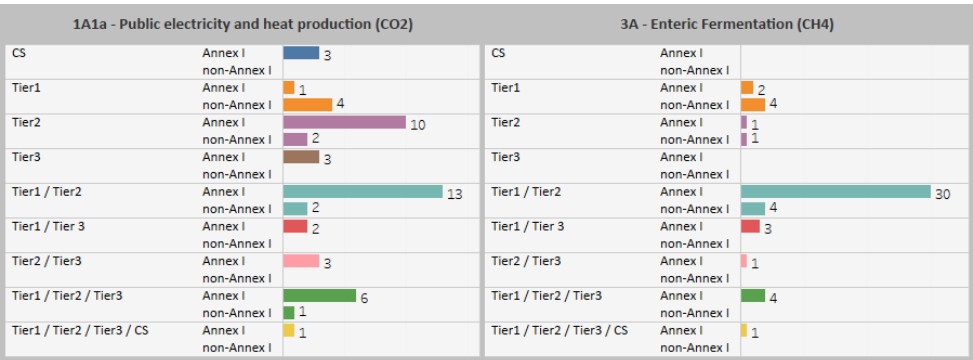

Figure 1. Number of G20 Annex-I and non-Annex I countries that reported to the UNFCCC inventory system by methodology applied for the 3A and 1A1a categories ($CH_4$ and $CO_2$ emissions) -2021

Source: UNFCCC Annex I and non-Annex I reports (last access May 2025),

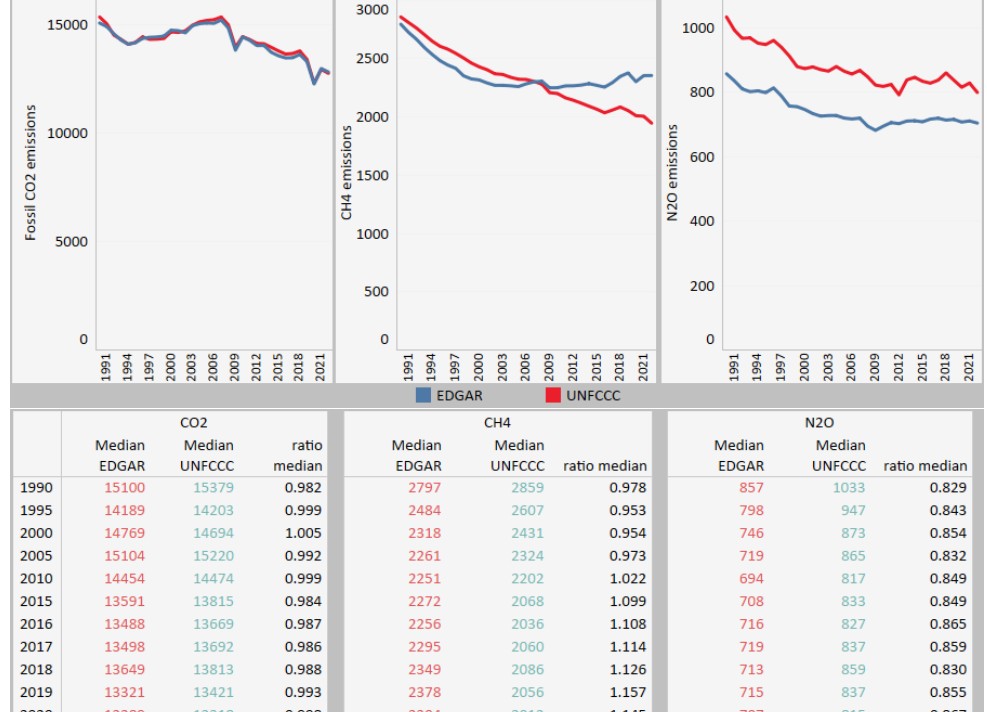

| | CO2 | | | CH4 | | | N2O | | |
|---|---|---|---|---|---|---|---|---|---|
| | Median EDGAR | Median UNFCCC | ratio median | Median EDGAR | Median UNFCCC | ratio median | Median EDGAR | Median UNFCCC | ratio median |
| 1990 | 15100 | 15379 | 0.982 | 2797 | 2859 | 0.978 | 857 | 1033 | 0.829 |
| 1995 | 14189 | 14203 | 0.999 | 2484 | 2607 | 0.953 | 798 | 947 | 0.843 |
| 2000 | 14769 | 14694 | 1.005 | 2318 | 2431 | 0.954 | 746 | 873 | 0.854 |
| 2005 | 15104 | 15220 | 0.992 | 2261 | 2324 | 0.973 | 719 | 865 | 0.832 |
| 2010 | 14454 | 14474 | 0.999 | 2251 | 2202 | 1.022 | 694 | 817 | 0.849 |
| 2015 | 13591 | 13815 | 0.984 | 2272 | 2068 | 1.099 | 708 | 833 | 0.849 |
| 2016 | 13488 | 13669 | 0.987 | 2256 | 2036 | 1.108 | 716 | 827 | 0.865 |
| 2017 | 13498 | 13692 | 0.986 | 2295 | 2060 | 1.114 | 719 | 837 | 0.859 |
| 2018 | 13649 | 13813 | 0.988 | 2349 | 2086 | 1.126 | 713 | 859 | 0.830 |
| 2019 | 13321 | 13421 | 0.993 | 2378 | 2056 | 1.157 | 715 | 837 | 0.855 |
| 2020 | 12289 | 12318 | 0.998 | 2304 | 2012 | 1.145 | 707 | 815 | 0.867 |
| 2021 | 12997 | 12961 | 1.003 | 2356 | 2007 | 1.174 | 710 | 828 | 0.858 |
| 2022 | 12842 | 12780 | 1.005 | 2356 | 1947 | 1.210 | 704 | 799 | 0.881 |

Figure 2. Temporal trend of fossil $CO_2$, $CH_4$ and $N_2O$ emissions for Annex I (above) – median values for EDGAR and UNFCCC Annex I and respective medians ratio for the selected years (below), 1990-2022, Mt $CO_2$-eq

Source: UNFCCC CRT 2024; EDGAR 2024



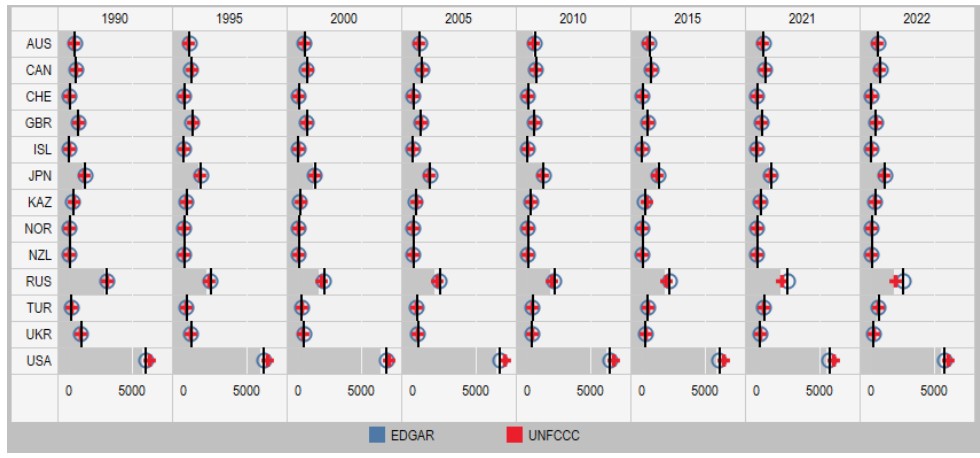

Figure 3. GHG ($CO_2$, $CH_4$ and $N_2O$) emissions every 5-years, 1990-2022, Annex I countries (EU27 not included), Mt $CO_2$-eq

Source: UNFCCC CRT 2024; EDGAR 2024

NB. GWP (100 years) of IPCC Fourth Assessment Report is applied for $CH_4$ and $N_2O$ emissions. EDGAR (blue circles) and UNFCCC submissions (red crosses). Highlighted area represents up to 90% of UNFCCC EU27 emissions.

Vertical line represents the median value of EDGAR emissions

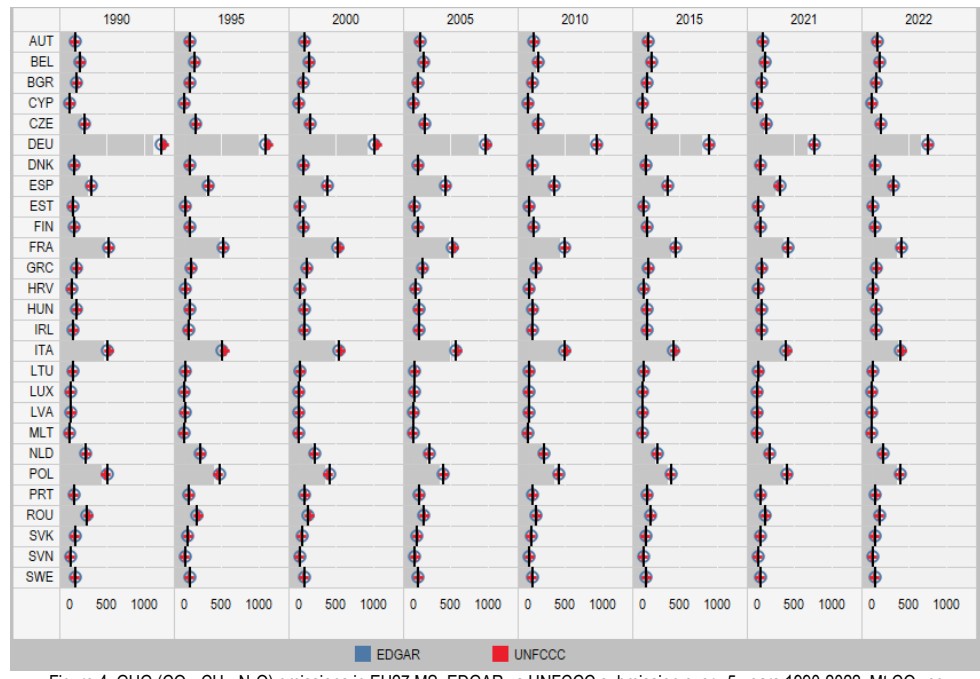

Figure 4. GHG ($CO_2$, $CH_4$, $N_2O$) emissions in EU27 MS: EDGAR vs UNFCCC submission every 5-years,1990-2022, Mt $CO_2$-eq

Source: UNFCCC CRT 2024, EDGAR 2024

NB. EDGAR (blue circles) and UNFCCC submissions (red crosses). Highlighted area represents up to 90% of UNFCCC EU27 emissions.

Vertical line represents the median value of EDGAR emissions



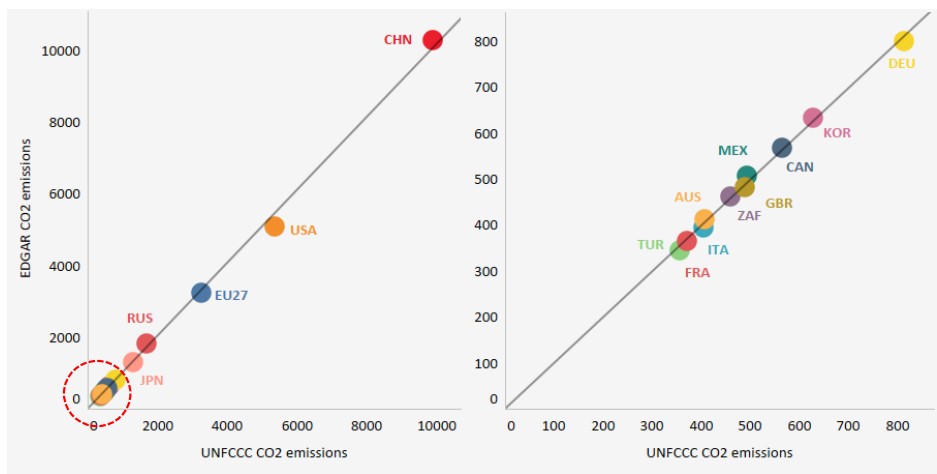

Figure 5. G20 countries fossil CO$_2$ emissions: EDGAR compared with UNFCCC, 2012, (Mt)

Source: UNFCCC CRT 2024, UNFCCC non-Annex I reports (last access May 2025), EDGAR 2024

NB. Countries with the largest CO$_2$ emissions, such as China (CHN), the USA, the EU27, and Russia (RUS), are positioned on the left side of the graph. In contrast, countries located in the lower-left section of the graph (inside the red circle) are displayed on the right side.

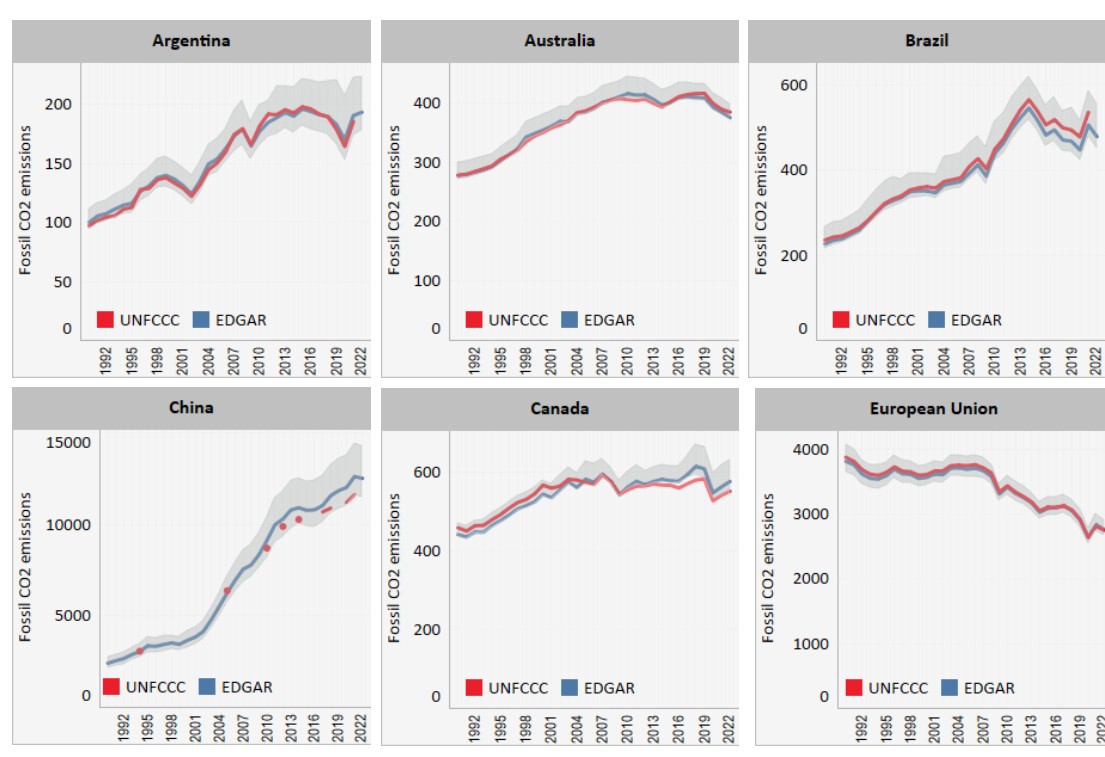





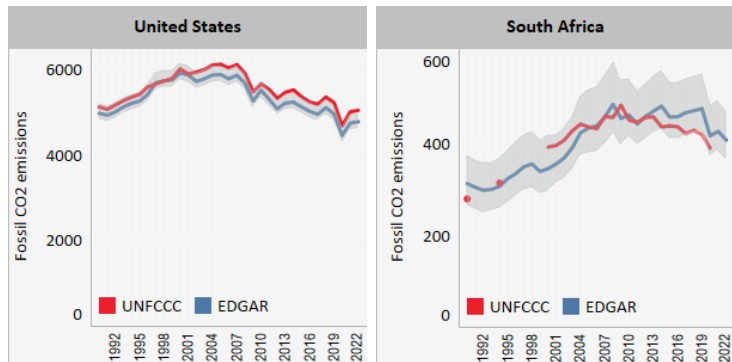

Figure 6. Temporal trends of fossil $CO_2$ emissions in G20 countries: EDGAR vs UNFCCC inventories, 1990-2022, Mt

Source: UNFCCC CRT 2024, EDGAR 2024, UNFCCC non-Annex I reports and CRT tables (last access May 2025)
NB: The shadow area represents the lower and the upper EDGAR emissions estimated uncertainty. The EDGARv8.0, 2023 dataset, incorporates or is consistent with the updated statistical data reported by Annex I countries in their 2023 submissions to the UNFCCC. For non-Annex I countries with submissions during year 2024 the EDGAR 2024 data are used for the comparison. The data for non-Annex I countries included here are China - the 2017 and 2018 data are sourced from the Second and Third Biennial Update Reports, submitted to the UNFCCC in December 2018 and 2023, respectively. 2020 and 2021 data are sourced from CRT tables submitted in December 2024. Brazil –data for period 1990-2021 are sourced from CRT tables submitted in December 2024. Argentina -data for period 1990-2021 are sourced from CRT tables submitted in December 2024. India – data are sourced from the 3rd and 4th NC submitted respectively in 2023 and 2024. Indonesia – data sourced from BURs (BUR3 submitted in 2021 but detailed data for gas for period 2000-2019 are missing). Mexico - data for period 2000-2015 are sourced from 2019 NC submission. South Africa - data for period 2000-2021 is sourced from the Biennial Transparency Report (BTR) submitted in December 2024. Saudi Arabia – data are sourced from BURs (BUR2 submitted April 2024). South Korea – data are sourced from BUR4 submitted in July 2023.

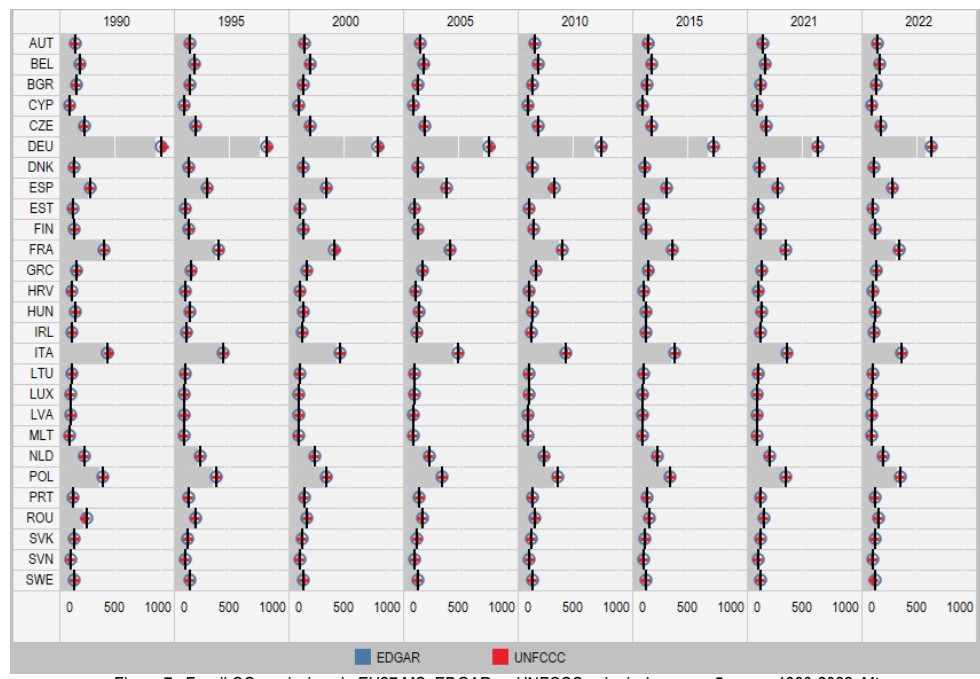

Figure 7.  Fossil $CO_2$ emissions in EU27 MS: EDGAR vs UNFCCC submission every 5-years, 1990-2022, Mt
Source: UNFCCC CRT 2024, EDGAR 2024
NB. EDGAR (blue circles) and UNFCCC submissions (red crosses). Highlighted area represents up to 90% of UNFCCC EU27 emissions.
Vertical line represents the median value of EDGAR emissions











Figure 8. Temporal trends of CH$_4$ emissions in G20 countries: EDGAR vs UNFCCC inventories, 1990-2021, Mt

Source: UNFCCC CRT 2024, EDGAR 2024, UNFCCC non-Annex I reports and CRT tables (last access May 2025)

NB: The shadow area represents the lower and the upper EDGAR emissions estimated uncertainty. The data for non-Annex I countries included here are: China - the 2017 and 2018 data are sourced from the Second and Third Biennial Update Reports, submitted to the UNFCCC in December 2018 and 2023, respectively. 2020 and 2021 data are sourced from CRT tables submitted in December 2024. Brazil –data for period 1990-2021 are sourced from CRT tables submitted in December 2024. Argentina -data for period 1990-2021 are sourced from CRT tables submitted in December 2024. India – data are sourced from the 3rd and 4th NC submitted respectively in 2023 and 2024. Indonesia – data sourced from BURs (BUR3 submitted in 2021 but detailed data for gas for period 2000-2019 are missing). Mexico -  data for period 2000-2015 are sourced from 2019 NC submission. South Africa - data for period 2000-2021 is sourced from the Biennial Transparency Report (BTR) submitted in December 2024. Saudi Arabia – data are sourced from BURs (BUR2 submitted April 2024). South Korea – data are sourced from BUR4 submitted in July 2023.



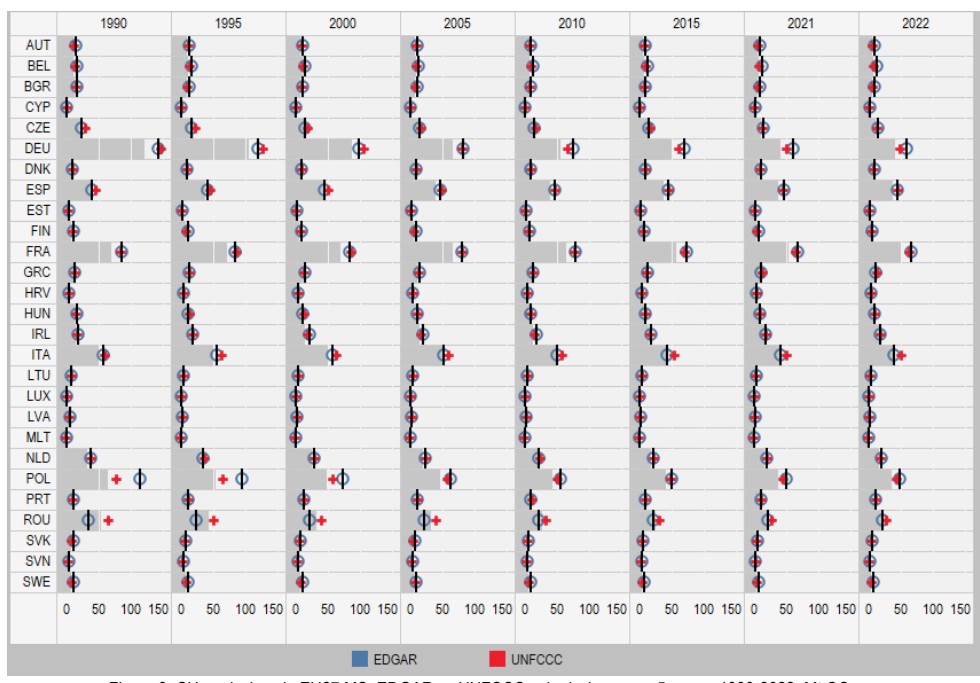

Figure 9. CH$_4$ emissions in EU27 MS: EDGAR vs UNFCCC submission every 5-years, 1990-2022, Mt CO$_2$-eq
EDGAR (blue circles) and UNFCCC submissions (red crosses). Highlighted area represents up to 90% of UNFCCC EU27 emissions.
Vertical line represents the median value of EDGAR emissions
Source: UNFCCC CRT 2024, EDGAR 2024









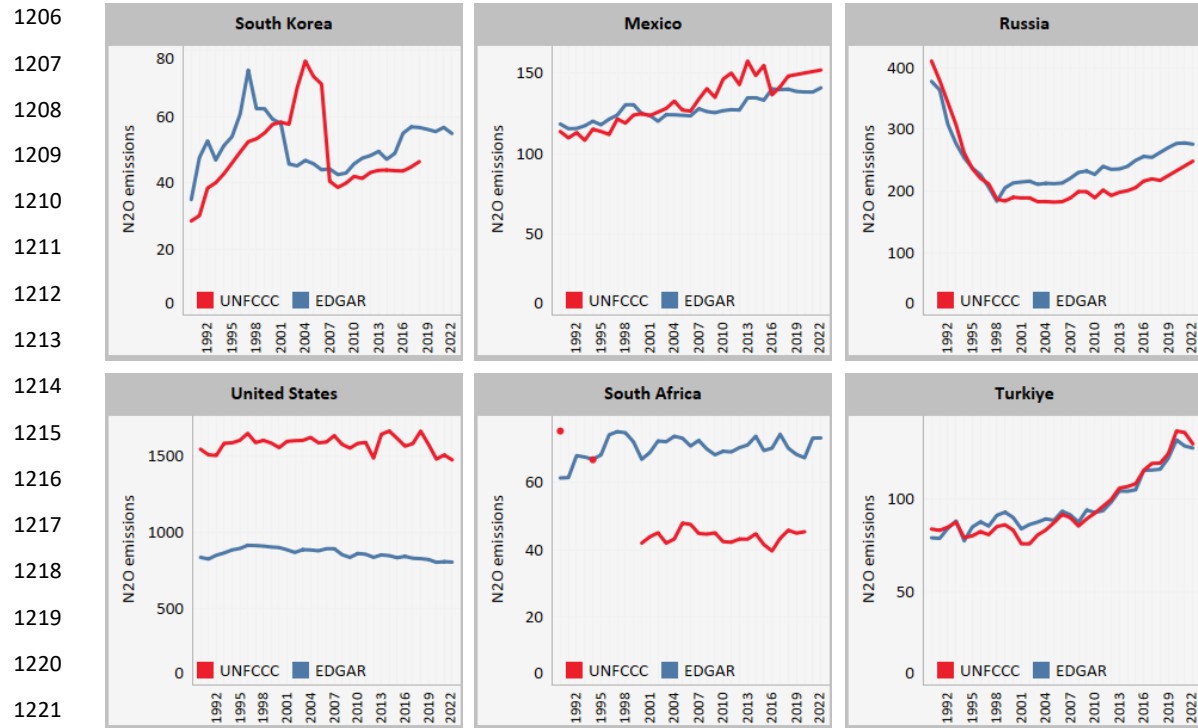

Figure 10. Temporal trend of N$_2$O emissions in G20 countries: EDGAR vs UNFCCC inventories, 1990-202, kt

Source: UNFCCC CRT 2024, EDGAR 2024, UNFCCC non-Annex I reports and CRT tables (last access May 2025)

NB: The shadow area represents the lower and the upper EDGAR emissions estimated uncertainty. The data for non-Annex I countries included here are: China - the 2017 and 2018 data are sourced from the Second and Third Biennial Update Reports, submitted to the UNFCCC in December 2018 and 2023, respectively. 2020 and 2021 data are sourced from CRT tables submitted in December 2024. Brazil –data for period 1990-2021 are sourced from CRT tables submitted in December 2024. Argentina -data for period 1990-2021 are sourced from CRT tables submitted in December 2024. India – data are sourced from the 3$^{rd}$ and 4$^{th}$ NC submitted respectively in 2023 and 2024. Indonesia – data sourced from BURs (BUR3 submitted in 2021 but detailed data for gas for period 2000-2019 are missing). Mexico - data for period 2000-2015 are sourced from 2019 NC submission. South Africa - data for period 2000-2021 is sourced from the Biennial Transparency Report (BTR) submitted in December 2024. Saudi Arabia – data are sourced from BURs (BUR2 submitted April 2024). South Korea – data are sourced from BUR4 submitted in July 2023.



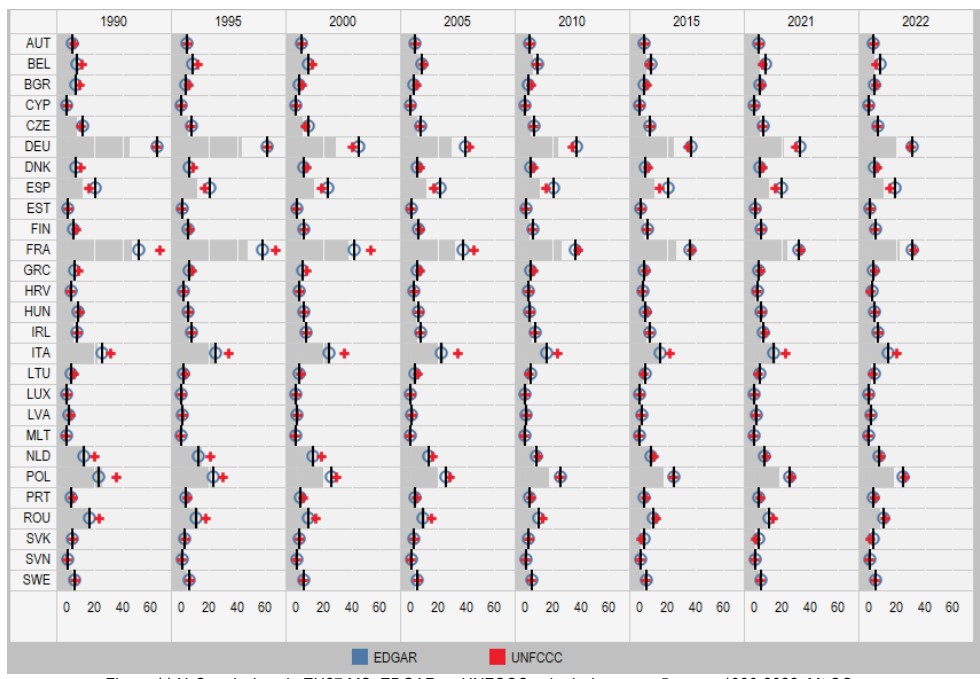

Figure 11.N₂O emissions in EU27 MS: EDGAR vs UNFCCC submission every 5-years, 1990-2022, Mt CO₂-eq
EDGAR (blue circles) and UNFCCC submissions (red crosses). Highlighted area represents up to 90% of UNFCCC EU27 emissions.
Vertical line represents the median value of EDGAR emissions
Source: UNFCCC CRT 2024, EDGAR 2024



**Table 1.** Key milestones in the UNFCCC inventory system reporting

| Year | Development | Implications |
|---|---|---|
| 1999 | Introduction of CRF tables | Standardized reporting format for Annex I Parties, enabling comparability |
| 2014 | Launch of Biennial Update Reports (BURs) for Non-Annex I Parties | Non-Annex I Parties began submitting BURs, enhancing transparency while considering their capabilities. |
| 2015 | Paris Agreement Adoption | Establishment of the Enhanced Transparency Framework (ETF) to replace the existing MRV system and standardize reporting for all Parties. |
| 2023 | Introduction of Test CRTs for Feedback | Parties tested the CRTs and provided feedback for the final versions, aligning with the 2006 IPCC Guidelines |
| 2024 | Transition to CRTs for Annex I Parties | CRTs replace CRFs for GHG inventory reporting, and all Parties submit Biennial Transparency Reports (BTRs) under the ETF. |

Source: UNFCCC, last access May 2025
Table 2. Data availability of total GHG emissions (without LULUCF) in non-Annex I G20 countries, 1990-2021

| Argentina | Brazil | China | India | Indonesia | Mexico | South Korea | South Africa | Saudi Arabia |
|---|---|---|---|---|---|---|---|---|
| 100% | 100% | 26% | 42% | 68% | 100% | 94% | 71% | 19% |

Source: UNFCCC last access May 2025, JRC elaboration
NB. The percentages included in this table indicate data availability, calculated as the ratio of the number of years a non-Annex I country
has reported data to the total number of years in the 1990–2021 period (31 years). These data are derived from non-Annex I countries
BURs, NCs and CRTs submitted to the UNFCCC. Data coverage elaborated using G20 non-Annex I countries' BURs, NCs and CRTs differs
from what is available on the UNFCCC webpage (country profiles and detailed data by parties). For Argentina, the data coverage on the
UNFCCC webpage corresponds to 19% coverage for period 1990-2021 whereas Argentina has now submitted its CRT for 1990-2022. For
China (CRT available only for 2005, 2020 and 2021), India and Saudi Arabia (CRT available only for 2019, 2020, 2021) the available data
on the UNFCCC website corresponds to 13% coverage for period 1990-2021.  For Indonesia, the available data on the UNFCCC website
corresponds to 20% coverage for the period 1990-2021. For South Africa, the available data on the UNFCCC website correspond to years
1990 and 1994 only. For Mexico the available data on the UNFCCC website covers only the period 1990-2013.
Table 3. Uncertainties in $CO_2$, $CH_4$ and $N_2O$ emissions estimates in EDGAR

| | Sector | Uncertainty (%) | Notes |
|---|---|---|---|
| $CO_2$ | Energy (fossil fuel combustion) | ±5–10% (industrialized countries); ±10–20% (developing countries) | Lower uncertainty due to robust activity data and emission factors |
| $CH_4$ | Energy (fugitive emissions), Agriculture, Waste | ±50–150% (depending on source and region) | High variability due to spatial, process, and reporting differences |
| $N_2O$ | Agriculture, Fossil fuel combustion, Waste | ±50–100% (fossil fuel combustion); >100% (agriculture, waste) | Significant uncertainty from complex chemical/biological processes |

Source: Crippa et al., (2024) based on Solazzo et al., (2021) methodology





Table 4. GHG emissions (CO$_2$, CH$_4$, N$_2$O) in G20: EDGAR vs UNFCCC submissions: relative differences over years, 1990-2022, and
uncertainties: (EDGAR average 1990-2022; UNFCCC from 2022 submissions where available) (%)

| | 1990 | 2000 | 2005 | 2010 | 2012 | 2015 | 2016 | 2017 | 2018 | 2019 | 2020 | 2021 | 2022 | UNFCCC | EDGAR |
|---|---|---|---|---|---|---|---|---|---|---|---|---|---|---|---|
| ARG | 14.91 | 6.25 | 7.95 | 1.59 | 2.58 | 2.90 | 3.17 | 4.45 | 3.69 | 4.38 | 5.27 | 2.88 | | 4.7 | 30.2 |
| AUS | 5.00 | 8.24 | 7.67 | 10.29 | 10.93 | 10.28 | 8.92 | 9.36 | 8.63 | 9.39 | 8.59 | 9.76 | 8.68 | 3.5 | 13.9 |
| BRA | -1.73 | 3.72 | 3.42 | 4.35 | 5.33 | 4.68 | 4.52 | 4.34 | 4.07 | 4.45 | 4.15 | 4.26 | | 20 | 32.6 |
| CAN | -5.14 | -6.57 | -2.77 | -2.02 | -2.13 | -0.08 | 1.10 | 2.33 | 4.08 | 3.36 | 2.38 | 2.91 | 3.98 | 2.6 | 8.9 |
| CHN | | | -0.98 | 3.79 | 3.25 | | 16.43 | | 19.32 | | 4.92 | 6.06 | | 4.1-4.4 | 14.0 |
| EU27 | 0.15 | 0.18 | -0.18 | 1.51 | 1.68 | 1.84 | 2.25 | 2.25 | 2.24 | 2.67 | 2.93 | 3.29 | 3.17 | 3.1 | 7.1 |
| DEU | -1.53 | -1.29 | -1.07 | 1.45 | 1.00 | 0.78 | 1.58 | 1.45 | 1.05 | 1.36 | 2.31 | 3.00 | 1.42 | 3.5 | 6.8 |
| FRA | -0.66 | -1.08 | -1.40 | 1.35 | 3.16 | 1.90 | 1.80 | 1.91 | 3.35 | 3.33 | 2.63 | 4.73 | 5.66 | 6.2 | 9.6 |
| GBR | -6.09 | -5.19 | -4.33 | -4.25 | -3.17 | -2.66 | -1.92 | -2.02 | -2.46 | -3.18 | -2.76 | -2.01 | -1.59 | 2.6 | 6.7 |
| IDN | 30.98 | -11.57 | | | | | | | | | | | | - | 30.2 |
| IND | | 11.93 | | 22.36 | | | 13.93 | | | | 10.96 | | | 6.85 | 19.3 |
| ITA | -1.62 | -3.29 | -2.55 | -3.39 | -3.34 | -3.50 | -4.12 | -3.31 | -3.61 | -3.25 | -4.15 | -2.55 | -3.76 | 2.4 | 5.4 |
| JPN | 3.61 | 0.56 | 1.22 | 2.82 | 2.49 | 3.08 | 4.51 | 4.61 | 5.28 | 5.22 | 5.30 | 4.43 | 0.22 | (-2.5 +2) | 6.6 |
| KOR | 7.59 | 6.08 | 1.68 | 0.40 | 0.77 | 1.58 | 3.87 | 3.12 | 1.31 | | | | | - | 6.7 |
| MEX | -3.34 | -3.20 | -1.65 | -5.92 | -2.09 | -6.05 | -9.82 | -5.47 | -8.72 | -7.17 | -9.77 | -5.72 | -8.88 | 7.5 | 9.1 |
| RUS | -2.98 | 9.99 | 10.04 | 9.11 | 10.09 | 12.36 | 12.01 | 11.98 | 13.56 | 18.04 | 18.30 | 19.97 | 26.93 | 12 | 11.1 |
| SAU | 36.61 | 11.88 | | 7.22 | 14.88 | | 7.95 | | | 2.95 | -1.23 | -2.88 | | - | 31.7 |
| TUR | -2.88 | 2.32 | -5.17 | 2.17 | 0.67 | 3.37 | 4.19 | 7.53 | 9.23 | 9.90 | 10.04 | 9.21 | 7.81 | 5.5 | 8.7 |
| USA | -5.72 | -4.07 | -6.23 | -5.71 | -6.65 | -6.79 | -6.41 | -6.85 | -6.86 | -6.47 | -6.32 | -6.04 | -5.50 | (-2 +6) | 6.3 |
| ZAF | 12.53 | -5.15 | 5.11 | 8.04 | 6.58 | 10.34 | 10.81 | 16.28 | 15.07 | 18.04 | 11.97 | | | (-5.7 +6... | 17.1 |

Source: UNFCCC CRT 2024, UNFCCC non-Annex I reports and CRT tables (last access May 2025), EDGAR 2024
NB. Empty cells indicate that data are missing in the UNFCCC country submission. The analysis for EU27 MS is shown in Table 5. GHG
emissions in the table represent CO$_2$, CH$_4$, and N$_2$O, expressed in kt CO$_2$-eq using IPCC fifth Assessment Report GWP values for all
countries. The EDGAR 2024 dataset incorporates or is consistent with the updated statistical data reported by Annex I countries in their
2024 submissions to the UNFCCC. For non-Annex I countries with submissions during year 2024 the EDGAR 2024 data are used for the
comparison. The data for non-Annex I countries included here are China - the 2017 and 2018 data are sourced from the Second and Third
Biennial Update Reports, submitted to the UNFCCC in December 2018 and 2023, respectively. The 2020 and 2021 data are sourced from
CRT tables submitted in December 2024. Brazil –data for period 1990-2021 are sourced from CRT tables submitted in December 2024.
Argentina -data for period 1990-2021 are sourced from CRT tables submitted in December 2024. India – data are sourced from the 3rd and
4th NC submitted respectively in 2023 and 2024. Indonesia – data sourced from BURs (BUR3 submitted in 2021 but detailed data for gas
for period 2000-2019 are missing). Mexico - data for period 1990-2022 are sourced from 2024 BTR submission. South Africa - data for
period 2000-2020 is sourced from the Biennial Transparency Report (BTR) submitted in December 2024. Saudi Arabia – data are sourced
from BURs (BUR2 submitted April 2024). South Korea – data are sourced from BUR4 submitted in July 2023.





*Table 5. GHG emissions in EU27 MS: EDGAR vs UNFCCC submissions: relative differences over years, 1990-2021 (%)*

|  | 1990 | 2000 | 2005 | 2010 | 2012 | 2015 | 2016 | 2017 | 2018 | 2019 | 2020 | 2021 | 2022 |
|---|---|---|---|---|---|---|---|---|---|---|---|---|---|
| AUT | 1.74 | 3.56 | 3.88 | 4.88 | 4.93 | 4.10 | 3.57 | 3.61 | 4.19 | 4.25 | 4.38 | 4.35 | 3.52 |
| BEL | -3.63 | 1.56 | -1.63 | 4.56 | 5.76 | 6.40 | 6.99 | 5.72 | 5.44 | 6.03 | 7.82 | 8.67 | 8.40 |
| BGR | 4.69 | 8.57 | 5.96 | 5.01 | 4.97 | 5.66 | 6.16 | 6.90 | 8.68 | 8.31 | 11.68 | 11.10 | 9.95 |
| CYP | -2.75 | -2.14 | -2.59 | 0.00 | -0.19 | 0.26 | 0.29 | 0.64 | 4.46 | 9.69 | 9.58 | 10.29 | 14.87 |
| CZE | 0.30 | 4.72 | 2.78 | 3.42 | 2.50 | 3.30 | 2.94 | 2.93 | 3.23 | 3.17 | 4.04 | 3.92 | 5.85 |
| DEU | -1.18 | -0.49 | -0.39 | 1.57 | 1.12 | 0.90 | 1.71 | 1.60 | 1.20 | 1.51 | 2.47 | 3.14 | 1.60 |
| DNK | -1.90 | -2.44 | -1.63 | -1.43 | -2.55 | -0.20 | -0.32 | -0.08 | 0.74 | 2.09 | 3.09 | 3.92 | 3.49 |
| ESP | 3.68 | 3.53 | 3.44 | 5.86 | 5.94 | 5.39 | 6.60 | 6.51 | 6.24 | 6.76 | 7.55 | 7.20 | 3.84 |
| EST | 10.40 | 18.55 | 17.79 | 18.94 | 21.57 | 33.52 | 21.75 | 21.27 | 32.80 | 29.45 | 27.93 | 21.50 | 11.59 |
| FIN | 1.49 | 5.15 | 5.89 | 7.96 | 8.01 | 9.69 | 9.10 | 9.11 | 9.73 | 9.97 | 10.63 | 10.31 | 5.63 |
| FRA | -0.90 | -0.61 | -1.00 | 1.12 | 2.78 | 1.67 | 1.58 | 1.68 | 3.10 | 3.14 | 2.49 | 4.55 | 5.39 |
| GRC | -4.74 | -4.42 | -6.26 | -6.44 | -8.13 | -4.76 | -2.49 | -6.03 | -5.51 | -4.81 | -5.47 | -6.33 | -6.59 |
| HRV | 6.03 | -3.15 | -3.44 | -2.84 | -2.91 | 1.97 | 3.50 | 2.57 | 2.92 | 3.27 | 4.71 | 3.81 | 1.79 |
| HUN | 1.78 | 2.70 | 3.16 | 1.77 | 2.74 | 2.15 | 2.35 | 3.47 | 3.53 | 3.98 | 5.04 | 5.28 | 7.95 |
| IRL | 3.66 | 7.34 | 6.74 | 6.19 | 6.80 | 5.77 | 7.08 | 6.58 | 0.02 | 0.95 | 0.87 | -0.71 | -0.37 |
| ITA | -1.80 | -2.98 | -2.37 | -3.10 | -2.81 | -2.73 | -3.12 | -2.10 | -2.35 | -1.75 | -2.35 | -0.94 | -2.23 |
| LTU | -2.73 | -4.92 | -3.02 | 10.14 | 14.14 | 16.23 | 13.02 | 14.14 | 14.58 | 14.39 | 13.35 | 11.51 | 10.05 |
| LUX | 1.49 | 1.62 | 0.45 | 0.35 | -0.02 | -0.69 | -0.85 | -1.11 | -1.42 | -1.59 | -2.22 | -1.87 | -0.81 |
| LVA | 4.08 | 0.25 | 5.92 | 6.07 | 10.70 | 11.24 | 11.53 | 10.07 | 10.74 | 9.98 | 11.03 | 10.53 | 9.52 |
| MLT | -5.16 | -14.77 | -3.46 | -0.13 | -2.23 | -4.69 | -5.10 | -2.94 | -3.87 | -4.24 | -5.16 | -5.38 | -6.63 |
| NLD | 0.60 | 2.65 | 3.12 | 2.46 | 4.00 | 3.74 | 4.49 | 4.02 | 3.57 | 3.79 | 3.94 | 4.89 | 5.39 |
| POL | 7.86 | 5.21 | 2.50 | 2.54 | 2.81 | 1.49 | 1.59 | 1.34 | 1.12 | 2.26 | 1.78 | 1.55 | 4.51 |
| PRT | -0.57 | -1.49 | -3.02 | -0.63 | -1.30 | -1.20 | 0.21 | 0.46 | 0.20 | -0.84 | -0.08 | -0.59 | -0.07 |
| ROU | -5.96 | -7.97 | -7.64 | -5.92 | -6.63 | -2.29 | -0.70 | -0.22 | 0.31 | 1.54 | -0.75 | 0.28 | 0.90 |
| SVK | 2.56 | 5.43 | 3.18 | 8.72 | 8.03 | 8.73 | 9.88 | 12.24 | 11.62 | 11.74 | 14.50 | 13.65 | 22.89 |
| SVN | 13.96 | 8.97 | 14.33 | 11.73 | 7.00 | 11.70 | 10.27 | 11.80 | 12.14 | 11.57 | 11.90 | 11.80 | 9.92 |
| SWE | 5.38 | 11.01 | 7.02 | 6.33 | 6.89 | 7.59 | 7.42 | 7.86 | 4.10 | 6.29 | 13.14 | 13.00 | 13.59 |
| EU27 | 0.38 | 0.81 | 0.36 | 1.67 | 1.85 | 1.98 | 2.37 | 2.38 | 2.36 | 2.78 | 3.07 | 3.57 | 3.29 |

Source: UNFCCC CRT 2024, EDGAR 2024
NB: IPCC GWP (100 years) AR5 values are used in both 2023 EU27 countries submissions and EDGARv8.0.
Table 6. Fossil $CO_2$ emissions in G20: EDGAR vs UNFCCC submissions: relative differences over years, 1990-2022 (%)

|  | 1990 | 2000 | 2005 | 2010 | 2012 | 2015 | 2016 | 2017 | 2018 | 2019 | 2020 | 2021 | 2022 |
|---|---|---|---|---|---|---|---|---|---|---|---|---|---|
| ARG | 2,22 | 2,03 | 1,60 | -3,57 | -2,91 | -2,15 | -2,50 | -1,78 | -1,83 | -0,91 | -0,27 | -2,22 |  |
| AUS | -0,16 | 1,10 | -0,54 | 2,50 | 1,75 | -0,24 | -0,46 | -0,72 | -1,54 | -1,90 | -1,67 | -1,12 | -2,47 |
| BRA | -3,82 | -1,17 | -2,17 | -2,20 | -2,15 | -3,96 | -4,75 | -4,61 | -5,72 | -5,32 | -6,40 | -5,57 |  |
| CAN | -3,83 | -4,10 | 1,44 | 1,24 | 0,54 | 2,03 | 3,04 | 4,12 | 5,97 | 4,33 | 3,82 | 3,94 | 4,49 |
| CHN |  |  | -1,82 | 4,90 | 4,09 |  | 3,26 | 6,21 |  |  | 7,33 | 8,54 |  |
| DEU | -3,96 | -3,03 | -2,97 | -1,37 | -1,67 | -1,93 | -1,04 | -1,23 | -1,69 | -1,81 | -0,90 | -0,14 | -1,78 |
| EU27 | -1,63 | -1,19 | -1,44 | -0,56 | -0,47 | -0,50 | -0,14 | -0,13 | -0,19 | -0,01 | 0,43 | 0,91 | 0,60 |
| FRA | -2,95 | -2,72 | -3,67 | -1,99 | 0,19 | -2,20 | -2,22 | -1,95 | -0,69 | -0,46 | -0,48 | 1,72 | 2,88 |
| GBR | -3,53 | -3,30 | -2,21 | -2,58 | -1,62 | -1,87 | -1,45 | -1,59 | -2,15 | -2,85 | -2,78 | -2,02 | -1,82 |
| IDN | 13,38 | 3,30 |  |  |  |  |  |  |  | -6,67 |  |  |  |
| IND |  | -2,84 |  | 10,76 |  |  | 3,24 |  |  | 1,73 | -2,67 |  |  |
| ITA | -2,69 | -3,17 | -1,57 | -2,68 | -2,13 | -2,00 | -2,43 | -1,91 | -2,42 | -2,11 | -2,23 | -0,49 | -2,39 |
| JPN | 0,82 | -1,20 | -0,57 | 0,47 | 0,41 | 1,00 | 2,47 | 2,55 | 3,21 | 3,08 | 3,05 | 2,17 | -2,40 |
| KOR | 7,98 | 6,87 | 3,60 | 0,47 | 0,69 | 1,32 | 3,53 | 2,72 | 0,88 |  |  |  |  |
| MEX | -6,33 | -2,66 | -0,18 | -3,05 | 0,26 | -4,41 | -10,15 | -4,70 | -8,62 | -6,40 | -7,20 | -0,15 | -6,96 |
| RUS | -3,88 | 13,74 | 11,36 | 7,02 | 7,05 | 7,41 | 6,30 | 6,05 | 7,65 | 12,17 | 12,23 | 14,42 | 22,08 |
| SAU | 25,77 | 3,32 |  | 3,32 | 8,12 |  | 1,11 |  |  | -4,86 | -8,59 | -10,11 |  |
| TUR | 0,66 | -2,28 | -7,84 | -1,70 | -3,73 | -4,79 | -3,32 | 0,89 | 1,60 | 2,31 | 2,03 | 1,19 | -1,74 |
| USA | -2,88 | -1,56 | -3,89 | -2,59 | -4,59 | -4,38 | -4,14 | -4,56 | -4,55 | -5,11 | -4,75 | -5,22 | -5,27 |
| ZAF | 11,85 | -11,92 | -0,46 | 2,49 | 0,37 | 4,39 | 4,93 | 10,91 | 9,91 | 13,54 | 6,85 |  |  |

Source: UNFCCC CRT 2024, UNFCCC non-Annex I reports and CRT tables (last access May 2025), EDGAR 2024.
NB. Empty cells indicate that data were missing in the UNFCCC country submissions. The EDGAR 2024 dataset incorporates or is
consistent with the updated statistical data reported by Annex I countries in their 2024 submissions to the UNFCCC. For non-Annex I
countries with submissions during year 2024 the EDGAR 2024 data are used for the comparison. The data for non-Annex I countries
included here are China - the 2017 and 2018 data are sourced from the Second and Third Biennial Update Reports, submitted to the
UNFCCC in December 2018 and 2023, respectively. 2020 and 2021 data are sourced from CRT tables submitted in December 2024.



Brazil –data for period 1990-2021 are sourced from CRT tables submitted in December 2024. Argentina -data for period 1990-2021 are
sourced from CRT tables submitted in December 2024. India – data are sourced from the 3rd and 4th NC submitted respectively in 2023
and 2024. Indonesia – data sourced from BURs (BUR3 submitted in 2021 but detailed data for gas for period 2000-2019 are missing).
Mexico -   data for period 1990-2022 are sourced from 2024 BTR submission. South Africa - data for period 2000-2020 is sourced from
the Biennial Transparency Report (BTR) submitted in December 2024. Saudi Arabia – data are sourced from BURs (BUR2 submitted April
2024). South Korea – data are sourced from BUR4 submitted in July 2023.



Table 7. CH₄ emissions in G20: EDGAR vs UNFCCC submissions: relative differences over years, 1990-2022 (%)

| | 1990 | 2000 | 2005 | 2010 | 2012 | 2015 | 2016 | 2017 | 2018 | 2019 | 2020 | 2021 | 2022 |
|---|---|---|---|---|---|---|---|---|---|---|---|---|---|
| ARG | 15,4 | 0,5 | 4,1 | -1,4 | 0,5 | -0,2 | 0,3 | 2,0 | 0,4 | 0,5 | 1,2 | -0,7 | |
| AUS | -1,8 | 8,1 | 13,3 | 18,5 | 20,9 | 23,6 | 19,7 | 21,6 | 22,4 | 28,2 | 22,7 | 23,0 | 21,7 |
| BRA | 0,4 | 7,1 | 8,8 | 11,7 | 14,8 | 16,0 | 15,7 | 15,7 | 15,9 | 16,1 | 15,9 | 15,2 | |
| CAN | -3,0 | -5,8 | -6,3 | -5,4 | -3,9 | 1,1 | 2,6 | 4,6 | 6,2 | 8,4 | 7,1 | 11,7 | |
| CHN | | | 1,1 | -2,5 | -1,5 | | | | | | -2,4 | -0,3 | |
| DEU | 5,8 | 1,1 | 11,7 | 23,7 | 22,0 | 22,7 | 24,1 | 25,6 | 26,5 | 29,7 | 29,0 | 30,6 | 30,8 |
| EU27 | 8,4 | 5,3 | 5,3 | 9,2 | 9,0 | 9,5 | 10,4 | 11,1 | 11,6 | 12,7 | 11,1 | 11,8 | 11,9 |
| FRA | 10,4 | 7,3 | 8,2 | 10,2 | 11,7 | 14,4 | 14,3 | 15,0 | 17,1 | 14,0 | 12,1 | 13,0 | 12,3 |
| GBR | -10,9 | -18,0 | -20,9 | -22,5 | -21,8 | -16,5 | -14,0 | -13,5 | -13,0 | -13,5 | -11,8 | -9,7 | -9,2 |
| IDN | 26,7 | -36,6 | | | | | | | | | | | |
| IND | | 24,5 | | 38,4 | | | 41,2 | | | | 51,5 | | |
| ITA | 4,9 | -0,2 | -5,3 | -4,8 | -8,1 | -11,3 | -12,2 | -9,8 | -9,0 | -7,8 | -11,4 | -11,7 | -12,0 |
| JPN | 71,3 | 57,0 | 61,5 | 90,4 | 91,1 | 89,9 | 88,6 | 89,4 | 89,1 | 88,9 | 89,2 | 88,4 | 94,5 |
| KOR | 2,3 | -1,9 | -4,6 | -3,4 | -1,3 | 2,9 | 2,8 | 2,4 | 2,1 | | | | |
| MEX | 2,8 | -5,6 | -5,9 | -12,6 | -7,4 | -9,1 | -11,4 | -8,8 | -9,8 | -9,9 | -17,2 | -18,7 | -14,3 |
| RUS | 3,2 | -7,7 | 2,7 | 18,7 | 25,5 | 40,1 | 45,2 | 47,0 | 47,8 | 52,6 | 53,7 | 53,7 | 59,2 |
| SAU | 137,1 | 86,5 | | 43,3 | 89,7 | | 87,4 | | | 91,6 | 74,5 | 74,3 | |
| TUR | -12,3 | 20,6 | 5,4 | 23,4 | 24,9 | 58,4 | 53,6 | 56,3 | 60,9 | 57,3 | 61,8 | 65,7 | 70,8 |
| USA | -3,6 | -3,5 | -4,0 | -7,0 | -2,0 | -0,5 | -0,4 | -0,2 | 1,8 | 6,9 | 4,7 | 10,6 | 15,0 |
| ZAF | 26,6 | 6,7 | 11,7 | 16,9 | 19,1 | 20,5 | 20,2 | 20,3 | 19,9 | 19,2 | 14,9 | | |

Source: UNFCCC CRT 2024, UNFCCC non-Annex I reports and CRT tables (last access May 2025), EDGAR 2024
NB. Empty cells indicate that data were missing in the UNFCCC country submissions. The EDGAR 2024 dataset incorporates or is
consistent with the updated statistical data reported by Annex I countries in their 2024 submissions to the UNFCCC. For non-Annex I
countries with submissions during year 2024 the EDGAR 2024 data are used for the comparison. The data for non-Annex I countries
included here are China - the 2017 and 2018 data are sourced from the Second and Third Biennial Update Reports, submitted to the
UNFCCC in December 2018 and 2023, respectively. 2020 and 2021 data are sourced from CRT tables submitted in December 2024.
Brazil –data for period 1990-2021 are sourced from CRT tables submitted in December 2024. Argentina -data for period 1990-2021 are
sourced from CRT tables submitted in December 2024. India – data are sourced from the 3rd and 4th NC submitted respectively in 2023
and 2024. Indonesia – data sourced from BURs (BUR3 submitted in 2021 but detailed data for gas for period 2000-2019 are missing).
Mexico -   data for period 1990-2022 are sourced from 2024 BTR submission. South Africa - data for period 2000-2020 is sourced from
the Biennial Transparency Report (BTR) submitted in December 2024. Saudi Arabia – data are sourced from BURs (BUR2 submitted April
2024). South Korea – data are sourced from BUR4 submitted in July 2023.





Table 8. N$_2$O emissions in G20: EDGAR vs UNFCCC submissions: relative differences over years, 1990-2022 (%)

| | 1990 | 2000 | 2005 | 2010 | 2012 | 2015 | 2016 | 2017 | 2018 | 2019 | 2020 | 2021 | 2022 |
|---|---|---|---|---|---|---|---|---|---|---|---|---|---|
| ARG | 91,7 | 70,8 | 75,3 | 58,7 | 60,6 | 63,4 | 63,1 | 64,3 | 60,1 | 55,9 | 55,1 | 49,7 | |
| AUS | 184,8 | 168,6 | 152,4 | 146,9 | 164,8 | 181,3 | 175,9 | 173,7 | 171,6 | 176,8 | 175,2 | 177,9 | 180,7 |
| BRA | -5,2 | 7,1 | -0,8 | -0,4 | -0,3 | -2,3 | -2,0 | -2,9 | -3,4 | -2,9 | -2,5 | 0,9 | |
| CAN | 9,1 | 1,4 | -6,5 | 9,5 | 17,4 | 15,5 | 11,9 | 20,0 | 18,0 | 22,1 | 16,8 | 14,9 | 12,4 |
| CHN | | | 5,9 | 3,2 | 1,1 | | | -14,1 | -12,9 | | -23,6 | -26,7 | |
| DEU | 30,0 | 40,8 | 20,6 | 38,2 | 35,8 | 34,7 | 33,2 | 31,7 | 30,4 | 35,7 | 35,9 | 39,3 | 36,7 |
| EU27 | 5,5 | 9,5 | 8,8 | 21,4 | 23,1 | 24,2 | 24,4 | 22,8 | 21,9 | 23,8 | 21,9 | 22,3 | 25,1 |
| FRA | 0,2 | -0,7 | 5,8 | 23,7 | 20,5 | 22,3 | 22,0 | 19,0 | 20,0 | 23,9 | 14,9 | 22,2 | 23,8 |
| GBR | -26,4 | 12,7 | 13,0 | 17,7 | 22,1 | 22,0 | 24,4 | 23,4 | 22,9 | 20,8 | 24,7 | 20,7 | 25,9 |
| IDN | 233,3 | 134,9 | | | | | | | | | | | |
| IND | | 132,3 | | 113,8 | | | 77,4 | | | | 88,3 | | |
| ITA | 3,0 | -12,1 | -15,6 | -16,1 | -16,6 | -12,7 | -15,9 | -13,8 | -12,9 | -13,8 | -17,9 | -17,7 | -9,7 |
| JPN | -1,0 | -4,1 | 1,4 | -6,5 | -6,5 | -7,6 | -6,5 | -7,0 | -7,3 | -5,9 | -6,3 | -4,3 | -5,4 |
| KOR | 22,7 | 2,6 | -36,6 | 9,1 | 11,8 | 12,0 | 26,2 | 27,0 | 22,3 | | | | |
| MEX | 4,1 | 0,1 | -2,6 | -13,4 | -11,0 | -13,9 | 2,7 | -1,2 | -5,5 | -5,1 | -3,0 | -4,7 | -7,3 |
| RUS | -8,0 | 12,2 | 16,3 | 20,0 | 21,9 | 21,4 | 18,3 | 15,8 | 20,8 | 20,8 | 19,5 | 15,7 | 11,1 |
| SAU | -43,4 | -41,9 | | -19,9 | -19,7 | | -13,0 | | | 13,1 | 25,9 | 23,6 | |
| TUR | -5,7 | 8,2 | 1,4 | 0,4 | -1,4 | -3,0 | -0,1 | -3,0 | -2,8 | -1,9 | -3,5 | -5,4 | -1,7 |
| USA | -45,9 | -42,1 | -44,6 | -45,6 | -43,7 | -48,4 | -46,1 | -47,5 | -50,2 | -47,8 | -45,7 | -46,4 | -45,4 |
| ZAF | -18,5 | 59,1 | 52,5 | 63,0 | 62,6 | 67,0 | 76,6 | 71,0 | 53,0 | 51,8 | 48,3 | | |


Source: UNFCCC CRT 2024, UNFCCC non-Annex I reports and CRT tables (last access May 2025), EDGAR 2024
NB. Empty cells indicate that data were missing in the UNFCCC country submissions. The data for non-Annex I countries included here
are: China - the 2017 and 2018 data are sourced from the Second and Third Biennial Update Reports, submitted to the UNFCCC in
December 2018 and 2023, respectively. 2020 and 2021 data are sourced from CRT tables submitted in December 2024. Brazil –data for
period 1990-2021 are sourced from CRT tables submitted in December 2024. Argentina -data for period 1990-2021 are sourced from CRT
tables submitted in December 2024. India – data are sourced from the 3rd and 4th NC submitted respectively in 2023 and 2024. Indonesia
– data sourced from BURs (BUR3 submitted in 2021 but detailed data for gas for period 2000-2019 are missing). Mexico -   data for period
2000-2015 are sourced from 2019 NC submission. South Africa - data for period 2000-2021 is sourced from the Biennial Transparency
Report (BTR) submitted in December 2024. Saudi Arabia – data are sourced from BURs (BUR2 submitted April 2024). South Korea –
data are sourced from BUR4 submitted in July 2023.