# Peer review of "A comparative analysis of EDGAR and UNFCCC GHG emissions 1"

_Earth System Science Data, 2025_

## Author Response (AR1)

**Review 1**

**General overview of the article**

This paper presents a comparison of greenhouse gas (GHG) emissions—by species type—between EDGAR estimates and G20 submissions to the UNFCCC. I believe this represents a valuable contribution to the existing literature, offering insights into the reasons behind discrepancies between global emission inventories and national submissions. This topic is of interest not only to the scientific community but also to national inventory compilers.

Furthermore, the paper highlights the importance of global emission inventories in providing a more comprehensive picture of GHG emissions, particularly for regions where data is not regularly updated or readily available.

With regard to comments and suggestions for improvement, at a high level, I would suggest enhancing the clarity of the Introduction. The sample of countries considered is occasionally not well explained, and the rationale for selecting certain sectors to emphasise differences (e.g. Figure 1 and the associated analysis, which refers to all Annex I countries, some of which are not part of the G20) is somewhat unclear.

We thank the reviewer for pointing out this. Yes Fig.1 include all Annex I countries and only G20 non-Annex I countries. See your answer at point 5.c of your comments/questions.

Another recurring issue in the Introduction is the assumption that readers are already familiar—or very familiar—with the methodology underpinning the EDGAR database. While this may be partially true, I believe the authors may be overestimating the extent to which the audience is acquainted with EDGAR's methodological details.

I would also recommend enriching Section 5 with further reflection on the limitations inherent to global emission inventories. In its current form, the manuscript rightly emphasises the need for such tools and the methodological improvements made to enhance the EDGAR database. However, there are intrinsic limitations associated with global coverage that could be acknowledged and discussed, thereby offering readers a more balanced and comprehensive perspective.

As noted earlier, the paper presents insightful findings and constitutes a meaningful contribution to ongoing discussions on best practices for emission reporting, quantification, and transparency frameworks.

Below, the authors will find specific comments relating to individual sections of the manuscript.

**Introduction**

1. I would encourage the authors to consider rearranging some of the paragraphs in the Introduction. At present, the opening section informs readers of the differences between EDGAR and national submissions to the UNFCCC. However, it assumes familiarity with the methodology, species, regions, and sectors covered by EDGAR. While these aspects are addressed later in the text, reordering the paragraph flow could improve readability.

We have elaborated slightly the Introduction section to assure a smooth readability and also included a sentence about EDGAR so the reader knows since the beginning the inventories that are subject of the comparison – ". One such global inventory is the Emissions Database for Global Atmospheric Research (EDGAR), which provides consistent global estimates of anthropogenic GHG, and air pollutant emissions based on international statistics and standardized methodologies (see more in Section 2.2).". Now the Introduction provides this flow:

Importance of GHG emissions quantification IPCC Guidelines evolution (UNFCCC, CFR/CRT) Short introduction of EDGAR Issues on data/methodology comparison Bottom-up inventories (strengths & weakness) Short literature review on inventories comparison Paper purpose and focus

2. Lines 131–140: Although this may be evident to the authors or to part of the audience, I would recommend elaborating on the tier-level differences between EDGAR and the UNFCCC dataset.

**We have elaborated this part of the text to provide more information**

3. Building on point 2: I cannot recall at present—are fuel carbon contents in EDGAR nationally or regionally representative? How does this compare with the UNFCCC dataset?

Inserted in the section of Methodology - EDGAR calculates emissions from fuel combustion using the default carbon contents and net calorific values provided in the IPCC Guidelines, which are globally averaged by fuel type. These defaults ensure consistency across countries but do not reflect country-specific fuel characteristics. UNFCCC submissions use nationally measured carbon contents and net calorific values for fuels in their NIR submissions. This allows more accurate  $\mathrm{CO}_2$  emission estimates that reflect local fuel quality and combustion characteristics, leading to potential differences when comparing national inventories with EDGAR estimates.4. Continuing from point 2: How does the EDGAR database currently account for differing assumptions regarding waste disposal (e.g. landfills) and leaks associated with natural

gas transport? Variations in emission modelling for these sectors could significantly contribute to observed discrepancies.

I appreciate that word count limitations and scope definitions may constrain the level of detail provided. Nonetheless, brief comments on the above aspects would help readers better understand the sources of divergence.

In the revised manuscript, we clarify in the Methods section that detailed explanations of EDGAR's waste and fugitive methodologies are provided later in the results (Sections 3.2 and 3.3) as well as in the yearly EDGAR publications. Waste methodology, including assumptions on landfilled quantities, composition, and recovery, is described in the CH4 section. Fugitive emissions are addressed in both the CO2 and CH4 sections, where EDGAR's use of satellite flaring data, steel production statistics, and UNFCCC/EPA datasets is explained.

- 5. I find Figure 1 somewhat confusing, for the following reasons:
- a. The paper aims to explore differences between EDGAR estimates and UNFCCC data for G20 countries. However, the figure appears to summarise methodologies for all Annex I countries. Is this correct?

Yes, Fig.1 shows methodologies for all Annex I countries - it is written in the text 42 countries. We have re-done the figure to include only G20 countries both Annex I and non Annex I

b. It is unclear why the authors have chosen to focus on these two sectors. Is this due to their proportional contribution to total GHG emissions across G20 countries?

We focus on two categories—public electricity and heat production (CO2, 1A1a) and enteric fermentation (CH4, 3A)—as they represent major sources of emissions in the G20 and highlight clear methodological contrasts.

c. Would it not be more informative to indicate the methodology/tier level used by each of the 20 countries for the selected sectors?

We thank the reviewer for this suggestion. Figure 1 was conceived as an example illustration of methodological diversity in two key sectors, not as a complete mapping of all G20 countries. To complement this, we have included a supplementary table showing more detailed methodologies for several sectors in a subset of Annex I countries, again with the purpose of illustrating diversification. We have re-done Figure 1 now only for G20 countries and modified the text

Figure 1 is an illustrative example, highlighting methodologies applied in two key sectors: public electricity and heat production (1.A.1.a,  $CO_2$ ) and enteric fermentation (3.A,  $CH_4$ ) for G20 Annex I (10 countries) and non-Annex I (9 countries) providing also a country-by-country mapping of tier applications for these two categories. To further illustrate the diversity of methodologies, Table S.2 provides more detailed information on several Annex I countries and additional sectors. Like Figure 1, this table serves as an example of methodological variation rather than an exhaustive review.

Figure 1 shows the reliance on Tier 1, Tier 2, and Tier 3 methodologies, as well as the use of country-specific (CS) emission factors, which vary considerably between the two sectors. In the public electricity and heat production sector, Tier 2 methodologies are predominantly used in Annex I countries, with four G20 countries applying this approach. Two G20 Annex I countries employ a combination of Tier 1 and Tier 2 methodologies, reflecting a moderate level of methodological refinement. More advanced approaches, such as the exclusive use of Tier 3 methods or a combination of Tier 1, Tier 2, and Tier 3, are applied by three G20 Annex I countries. Only one G20 country applies a country-specific methodology for this sector. In G20 non-Annex I countries, Tier 1 and Tier 1/Tier 2 methods are most common (six countries).

In contrast, the enteric fermentation sector primarily relies on simpler approaches. The combination of Tier 1 and Tier 2 methods is used by most G20 Annex I and non-Annex I countries (eight), indicating a preference for straightforward, less data-intensive estimation methods for methane emissions from livestock. Only one G20 Annex I country adopts a purely Tier 1 methodology. Advanced combinations, such as Tier 1, Tier 2, and Tier 3, are applied by four G20 Annex I countries. Among G20 non-Annex I countries, Tier 1 and Tier 1/Tier 2 methods are the most widely applied (eight countries).

d. When comparing methodologies, could the authors clarify which national communication cycle has been considered for the non-Annex I countries in the G20 bloc? And which submission year for the Annex I countries?

Submission of year 2023 is used and the reference year is 2021. Added in the Figure 1 data source: GHG Locator, 2023 submissions (last access May 2025)

6. Could the authors explain their rationale for presenting median figures across the sample? Why is this considered more relevant than reporting total emissions for the group of countries? Is this due to missing data for non-Annex I countries across the time series? If so, is the mean still representative?

Added a footnote related to the use of median in Fig.2 - Median values are shown to minimise the influence of outliers and reporting gaps in the G20 sample. Totals or means can be disproportionately affected by missing data for non-Annex I countries or by the very large contributions of a few economies (e.g., China, USA, India). The median therefore provides a more robust measure of the central tendency across the group.

**Global GHG emissions**

7. The title of this section may be misleading, as the analysis focuses on G20 countries rather than global emissions. Please consider revising the section title accordingly.- We thank the reviewer for this comment. To avoid misleading wording, we removed the term "Global" from the subsection titles in Section 3. As Section 3 is already introduced as focusing on the G20, we decided not to add "G20" to each subsection title to keep them concise while maintaining clarity.

**Global CO2 emissions**

- 8. A similar comment applies to this section title. Since the comparison centres on G20 countries, referring to it as "Global CO2 Emissions" may be inaccurate. removed Global (see comment in point 7)
- 9. The authors note systemic differences for Italy, Japan, and others. While the percentage differences are relatively small, have the authors explored the underlying causes? Could these be related to activity data or assumptions regarding fuel carbon content? Have the authors identified which sector contributes most to these differences?

We thank the reviewer for this comment. We investigated the case of Italy in more detail. The relative differences between EDGAR and the Italian national inventory remain within the uncertainty ranges reported by both datasets. The main drivers are (i) timing of updates, as EDGAR and the Italian inventory activity data for specific sectors are revised at different moments; (ii) methodological differences, with EDGAR applying mainly Tier 1 approaches while Italy applies higher-tier methods (T2 & T3); (iii) differences in activity data and sectoral allocation (as also illustrated in the 1.A.1.a example); and (iv) the use of country-specific emission factors in Italy, which can vary over time, compared to the more static regional defaults in EDGAR. These differences,

while systematic, remain within uncertainty bounds and do not affect overall emission trends.

10. When discussing fugitive emissions, could the authors specify which part of the natural gas or oil production value chain is responsible for the observed differences? It seems unusual that this would account for discrepancies in  $CO_2$  emissions, given that fugitive emissions are typically associated with methane. Are the authors referring instead to process-related  $CO_2$  emissions from industrial sources? Please clarify.

In Section 3.2 (when fugitive is mentioned for the first time) we have inserted a footnote to clarify that fugitive CO2 in EDGAR arises from process-related sources such as flaring and coke production, rather than leakage. These sources are estimated using international datasets (GGFR/NOAA, Worldsteel) as described in Crippa et al., 2025.

**Global CH4 emissions**

- 11. Again, the section title may be misleading given the focus on G20 countries. removed Global (see comment in point 7)
- 12. Lines 573–581: Could the authors comment on the role of assumptions regarding the quantity of waste landfilled, waste composition, and methane recovery from landfills? This text is added in Section 3.3: "In addition, EDGAR incorporates WB data on waste composition for specific reference years (2012 and 2018) applying interpolation or extrapolation where gaps exist. Landfill CH4 recovery rates are included where available from UNFCCC submissions. In contrast, UNFCCC inventories often rely on more detailed country-specific surveys of waste types, composition, and recovery efficiency, which explains part of the divergence."
- 13. I would also encourage the authors to briefly explain the FOD method used for methane estimation or provide references for readers to consult. Comments on model calibration would also be welcome.

We added a footnote to provide the reader on what consist of the FOD: The First Order Decay (FOD) model assumes that degradable organic carbon in landfilled waste decays gradually over time, generating methane with a time lag. Further details are provided in IPCC (2006).

**Global N2O emissions**

14. As with previous sections, the title may benefit from revision to reflect the actual scope of the analysis. – removed Global (see comment in point 7)

**Section 5**

15. I concur with the points raised by the authors. However, a deeper discussion of the limitations associated with global emission inventories—particularly the use of Tier 1 methodologies, the lack of local/national data on technology penetration, and the use of emission factors that may not be geographically representative—would be beneficial. Such additions would offer readers a more balanced and comprehensive understanding of the constraints inherent to global inventories.

We thank the reviewer for this useful suggestion. In the revised Discussion we have added a more detailed reflection on the limitations of global inventories. We now emphasize that EDGAR's reliance on Tier 1 methods, international statistics, and default emission factors implies limited representation of national technology penetration and geographically specific factors. We also highlight that EDGAR's consistency is influenced by revisions in its underlying datasets, which can introduce discontinuities. These points provide a more balanced perspective on the constraints inherent to global inventories, complementing our discussion of their value.

The insert at section 5.4 "A key limitation of global inventories such as EDGAR is their reliance on Tier 1 methodologies and default emission factors, which are applied consistently across all countries to ensure comparability. While this uniformity is a strength for global assessments, it also means that national circumstances—such as country-specific emission factors, technology penetration rates, or abatement practices—are often not reflected. National inventories, by contrast, can apply Tier 2 or Tier 3 approaches that incorporate more detailed activity data and locally representative emission factors. Another limitation is the dependence of EDGAR on international statistics (e.g., IEA, FAO, UN, WB), whose revisions or definitional changes may introduce discontinuities into the time series that are not present in national inventory recalculations. These aspects highlight that global inventories are best viewed as complementary to national inventories: they provide a consistent and independent reference across all countries but cannot substitute the granularity of country-specific reporting."

**Review 2**

**General comments**

The paper compares EDGAR inventory of three major greenhouse gases to the national inventory reports, sourced from UNFCCC database, countries National Communications and Biannual Update Reports. The comparison is useful to inventory compilers, as 2006 IPCC Inventory Guidelines explicitly mentioned EDGAR as a source of independent data suitable for checking the estimates produced for National Inventory Reports (transparency reports in recent wording). The comparison uses G20 subset, which is justified by covering majority of emissions for purposes like global stock take. The paper is well written and can be accepted with minor revision.

**Detailed comments**

- Suggest mentioning 2006 IPCC Inventory Guidelines (Vol.1, Chap 6) recommendation to use EDGAR and other global inventories as a part of the verification process. – inserted in the introduction
- 2. Need to mention some examples of emission uncertainties for non-G-20, non-Annex countries, that are revealed by differences of emission estimates by different NIR releases, especially for N2O and CH4.
  - For non-G20, non-Annex countries, emission uncertainties are reported in their NIR/NID when available, but these are beyond the scope of this study. In case the suggestion was for G20 non Annex I countries we extended the input to the section 2.5 of the paper including some more information about uncertainty for these countries non CO2 substances.
- 3. Would be interesting to have more details how "time series consistency" (term in 2006 IPCC Inventory Guidelines) is implemented in NIR/UNFCCC reports and EDGAR

In the Methods section we now explain how EDGAR ensures time series consistency through full recalculations whenever methods are revised, similar to national inventories. We also note that EDGAR's consistency is constrained by the international activity data it uses (IEA, FAO, UN, WB), which may cause step changes, while NIRs can adjust their own official statistics to maintain smoother trends.

4. Line 510-512 attributes source of activity data in EDGAR for fugitive emissions to IEA. The Table 1 below present the India's coal sector CH4 emissions, it shows a drop after 2012, while country statistics on coal use don't have exhibit such a drop. Can this change be attributed to IEA data or change in emission factors?

Table 1 EDGAR\_2024\_GHG estimates of India's fossil CH4 emissions from solid fuels (including coal mining, coal storage; numbers truncated not rounded)

| year   | 2009   | 2010   | 2011   | 2012   | 2013   | 2014   | 2015   | 2016   |
|--------|--------|--------|--------|--------|--------|--------|--------|--------|
| Gg CH4 | 2181.0 | 2187.6 | 2272.4 | 2391.8 | 1376.6 | 1533.4 | 1643.5 | 1690.3 |

We thank the reviewer for highlighting the case of India. We investigated this further and confirmed that the decline in EDGAR estimates for India's coal sector CH4 emissions after 2012 originates from changes in IEA coking coal activity data. The IEA World Balance Documentation (2023) explains that coking coal data for India are partly estimated by the IEA Secretariat based on trade partner statistics, due to a notable discrepancy between India's reported coal imports and the reported exports to India. These figures also include production of non-metallurgical coal reported by India, which does not fully align with IEA

coking coal definitions. This methodological inconsistency in the underlying activity data explains the observed drop in EDGAR estimates. We inserted a short sentence in the section 3.3 in the paragraph that tackle fugitive emissions discrepancies – "In the case of India, there is a decline in fugitive CH4 emissions from solid fuels after 2012 in EDGAR estimation, which is not detected in the official reporting. This discrepancy is due to a drop in the IEA coking coal statistics for India. India's coking coal figures are partly estimated by the IEA Secretariat because of large discrepancies between official reporting and trade statistics (IEA, 2023)."

**Technical corrections**

Line 59 References Pfenninger et al., 2014, Prina et al., 2020 not found in a reference list.

**References included in the list**

**References:**

Emissions Database for Global Atmospheric Research (EDGAR), release EDGAR\_2024\_GHG (1970 - 2023) of October 2024.

For this reference we use as following

EDGAR, 2024, 1970-2023: Crippa et al., 2024 JRC dataset, http://data.europa.eu/89h/88c4dde4-05e0-40cd-a5b9-19d536f1791a